# Score-based free-form architectures for high-dimensional Fokker-Planck equations

**Feng Liu**[1,3,4,5]  **Faguo Wu**[1,4,5,6,*]  **Xiao Zhang**[2,4,5,6,7,*]

[1]School of Artificial Intelligence, Beihang University
[2]School of Mathematical Sciences, Beihang University
[3]National Superior College for Engineers, Beihang University
[4]Key Laboratory of Mathematics, Informatics and Behavioral Semantics, MoE
[5]Zhongguancun Laboratory
[6]Beijing Advanced Innovation Center for Future Blockchain and Privacy Computing
[7]Hangzhou International Innovation Institute, Beihang University

## Abstract

Deep learning methods incorporate PDE residuals as the loss function for solving Fokker-Planck equations, and usually impose the proper normalization condition to avoid a trivial solution. However, soft constraints require careful balancing of multi-objective loss functions, and specific network architectures may limit representation capacity under hard constraints. In this paper, we propose a novel framework: *Fokker-Planck neural network (FPNN)* that adopts a score PDE loss to decouple the score learning and the density normalization into two stages. Our method allows free-form network architectures to model the unnormalized density and strictly satisfy normalization constraints by post-processing. We demonstrate the effectiveness on various high-dimensional steady-state Fokker-Planck (SFP) equations, achieving superior accuracy and over a $20\times$ speedup compared to state-of-the-art methods. Without any labeled data, FPNNs achieve the mean absolute percentage error (MAPE) of 11.36%, 13.87% and 12.72% for 4D Ring, 6D Unimodal and 6D Multi-modal problems respectively, requiring only 256, 980, and 980 parameters. Experimental results highlights the potential as a universal fast solver for handling more than 20-dimensional SFP equations, with great gains in efficiency, accuracy, memory and computational resource usage.

## 1 Introduction

The Fokker–Planck (FP) equation governs the time-varying response probability density function (PDF) of dynamical systems driven by stochastic processes (Risken & Caugheyz, 1991). It finds wide applications in statistical physics, chemistry, biology, mathematical finance, and structural dynamics (De Decker & Nicolis, 2020; Tu et al., 2020; Hu et al., 2021; Boghosian et al., 2017). Solving FP equations presents three main challenges: high-dimensional variables, unbounded spatial domains, and the normalization condition (NC). Grid-based methods face the curse of dimensionality, while path integral methods and Monte Carlo simulations are limited by noise and computational complexity (Naess & Moe, 2000; Elman et al., 2014; Chen & Majda, 2017; Natarajan et al., 2021).

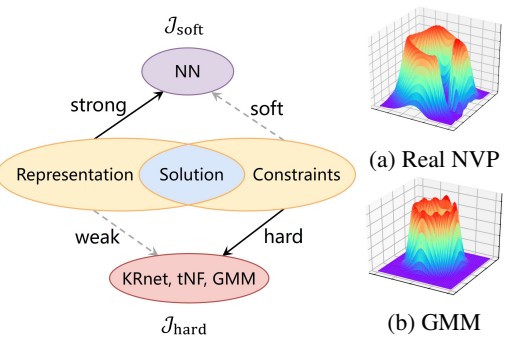

Figure 1: *Left*: Strengths and weaknesses for existing models. *Right*: Real NVP and GMM with 16 components fail to learn the ring function. The former consistently leaves a gap, while the latter exhibits numerous irregular protrusions.

---

*Corresponding author.
Email:{Liu_Feng,faguo,xiao.zh}@buaa.edu.cn

Deep learning methods, with their grid-free, causality-free, and feature-learning capabilities, have demonstrated potential in addressing high-dimensional partial differential equations (PDEs) (Han et al., 2018; Yu et al., 2018; Sirignano & Spiliopoulos, 2018; Liu et al., 2024). Among these promising methods, physics-informed neural networks (PINNs) (Raissi et al., 2019) leverage automatic differentiation to enforce the physical constraints of underlying PDEs and achieve great success in various problems, such as the Navier-Stokes equation, Burgers' equation, Schrödinger equation, etc. However, for steady-state Fokker-Planck (SFP) equations, PINN faces a new challenge: the zero solution also satisfies this equation and directly minimizing the plain PDE loss will quickly collapse the model to a trivial solution.

To mitigate this issue, data-driven methods guide the neural network (NN) to the desired solution by introducing labeled data or Kullback-Leibler divergence terms (Zhai et al., 2022; Chen et al., 2021). For 4D Ring problem, FP solver (Zhai et al., 2022) utilizes the direct Monte Carlo method with a very large number of particles ($10^{10}$ sample points) to obtain $10^4$ reference points. This method is computationally intensive and errors in the reference solutions may inversely limit the model's performance.

Table 1: Deep learning methods for SFP equations.

| Methods | Loss function | Model | Arbitrary NN? | Whether strictly satisfy NC? |
|---|---|---|---|---|
| Data-Driven | $\mathcal{J}_{\text{plain}} + \lambda \mathcal{J}_{\text{label}}$ | $p_\theta(\boldsymbol{x}) = p_{\text{NN}}(\boldsymbol{x}; \theta)$ | ✓ | ✗ |
| Normalization Condition | $\mathcal{J}_{\text{plain}} + \lambda \mathcal{J}_{\text{norm}}$ | $p_\theta(\boldsymbol{x}) = p_{\text{NN}}(\boldsymbol{x}; \theta)$ | ✓ | ✗ |
| | $\mathcal{J}_{\text{plain}}(p_\theta)$ | $p_\theta(\boldsymbol{x}) = p_{\text{KRnet}}(\boldsymbol{x}; \theta),\ p_{\text{GMM}}(\boldsymbol{x}; \theta),\ p_{\text{TNN}}(\boldsymbol{x}; \theta) \cdots$ | ✗ | ✓ |
| | $\mathcal{J}_{\text{score}}(\widetilde{p}_\theta)$ | $\widetilde{p}_\theta(\boldsymbol{x}) = \widetilde{p}_{\text{NN}}(\boldsymbol{x}; \theta),\ p_\theta(\boldsymbol{x}) = \frac{\widetilde{p}_\theta(\boldsymbol{x})}{\int \widetilde{p}_\theta(\boldsymbol{y})d\boldsymbol{y}}$ | ✓ | ✓ |

Another strategy that does not require labeled data is to impose normalization constraints in soft or hard manner. Soft constraints prevent zero solutions by adding a normalization penalty term that enforces the density integral to unity (Alhussein et al., 2023). While the normalization condition prevents the approximate solution from being zero across the entire domain, minimizing the plain PDE loss still has a tendency toward the trivial solution. These conflicting forces result in tortuous optimization dynamics (Wang et al., 2024; Al-Aradi et al., 2022), requiring delicate balancing of the multi-objective loss function. Specialized structures, such as normalizing flows (Dinh et al., 2016; Tang et al., 2022; Feng et al., 2022) and Gaussian mixture model (GMM) (Anderson & Farazmand, 2024), are developed to represent density functions and enforce hard constraints. These methods shrink the trial solution space and compromise the representation capability, leading to significant errors and computational burden when learning certain PDFs.

In summary, the former method requires manual balancing of multi-objective losses, while the latter may constrain the model's representation capacity. As shown in Figure 1, the ideal method strictly satisfies normalization constraints without sacrificing representation capacity. To the best of our knowledge, there is no effective approach to handle this conflict. In this paper, we propose a novel framework: Fokker-Planck Neural Network (FPNN) to decouple the fitting and normalizing stages through score PDE loss. FPNN efficiently solves high-dimensional SFP equations and offers the following advantages:

**Adaptive Domain for Complex Systems.** High-dimensional state spaces often exhibit intricate dynamics and interaction patterns. To accurately identify critical features and high-probability regions, we use the stochastic Runge–Kutta (SRK) method with strong order 1.5 to generate steady-state training data and adaptively determine the appropriate domain for SFP problems.

**Flexible Scaling for Small Solutions.** Since the probability density is non-negative and must integrate to unity over the spatial domain, the solution to high-dimensional problems is typically quite small and networks struggle to resolve such small yet meaningful scales directly. FPNN employs a score PDE loss, allowing the network to freely choose an appropriate scale and learn the unnormalized density. This effect of "data standardization" is infeasible for existing methods without labeled data or prior knowledge.

**Efficient Enforcement of Normalization Constraints.** The score loss allows free-form network architectures to strictly satisfy normalization constraints. FPNN avoids evaluating the density

function integral during training process and instead calculates the normalizing constant by post-processing.

**Scalability and Improvements in Optimization Dynamics.** Without any labeled data or the enforcement of zero boundary conditions, we successfully solve various 4-20 dimensional SFP equations for complex physical systems. FPNN circumvents the normalization condition in loss function which causes difficulty in training, and substantially improves optimization dynamics. Therefore, a large learning rate ($l = 0.01$) is allowed for fast descent and stable convergence, and FPNN achieves over $20\times$ speedup compared to state-of-the-art methods. We get the MAPE of 11.36%, 13.87% and 12.72% for 4D Ring, 6D Unimodal and 6D Multi-modal problems respectively, requiring only 256, 980, and 980 parameters. Our framework demonstrates substantial gains in efficiency, accuracy, and memory usage.

## 2 PRELIMINARIES

**Fokker–Planck equation.** Consider stochastic differential equations (SDEs) (Oksendal, 2013) of the form
$$d\mathbf{X} = \boldsymbol{\mu}(\mathbf{X})dt + \boldsymbol{\sigma}(\mathbf{X})d\mathbf{W}_t, \tag{1}$$
where the drift coefficient $\boldsymbol{\mu}(\mathbf{X}) \in \mathbb{R}^d$ is a vector field, $\boldsymbol{\sigma}(\mathbf{X}) \in \mathbb{R}^{d \times M}$ is a matrix-valued function and $\mathbf{W}_t$ is an $M$-dimensional standard Wiener process. The density function $p(\boldsymbol{x})$[1] of state variable $\mathbf{X}$ represents an invariant distribution, satisfying the steady-state Fokker-Planck (SFP) equation:

$$\frac{\partial p(\boldsymbol{x})}{\partial t} = \mathcal{L}p := -\sum_{i=1}^{d} \frac{\partial(p\mu_i)}{\partial x_i} + \sum_{i=1}^{d}\sum_{j=1}^{d} \frac{\partial^2(D_{i,j}p)}{\partial x_i \partial x_j} = -\nabla \cdot (p\boldsymbol{\mu}) + \nabla \cdot [\nabla \cdot (\boldsymbol{D}p)], \tag{2}$$

$$i.e. \quad \mathcal{L}p(\boldsymbol{x}) = 0, \qquad p(\boldsymbol{x}) \to 0 \quad (||\boldsymbol{x}|| \to \infty), \qquad \int_{\mathbb{R}^d} p(\boldsymbol{x})d\boldsymbol{x} = 1,$$

where $\boldsymbol{x} \in \mathbb{R}^d$ is the spatial variable, $\boldsymbol{D}(\boldsymbol{x}) = \frac{1}{2}\boldsymbol{\sigma}(\boldsymbol{x})\boldsymbol{\sigma}(\boldsymbol{x})^T$ is the diffusion matrix, $\mathcal{L}$ denotes the Fokker-Planck differential operator and $||\boldsymbol{x}||$ indicates the $\ell_2$ norm of $\boldsymbol{x}$.

**Physics-informed neural networks.** PINN is a deep neural network $p_\theta(\boldsymbol{x})$ to approximate the solution $p(\boldsymbol{x})$ with the universal approximation theorem (Hornik et al., 1989). The SFP equation in Eq. (2) can be expressed as,
$$\mathcal{L}p(\boldsymbol{x}) = 0 \quad \boldsymbol{x} \in \Omega, \qquad \mathcal{B}_i p(\boldsymbol{x}_i) = 0 \quad \boldsymbol{x}_i \in \partial\Omega_i, \qquad \mathcal{I}p = 1. \tag{3}$$
Here, $\mathcal{B}_i$ are boundary conditions (BCs) for boundaries $\partial\Omega_i \subset \Omega$, and the integral operator $\mathcal{I}$ represents the NC. These physical constraints are incorporated as regularization (typically mean-squared error terms) in the loss function. To accelerate model training and reduce violations of physical laws, hard constraints can be used to omit part loss terms.

**Score-based generative model.** Score matching (Hyvärinen & Dayan, 2005) is a popular method for learning unnormalized statistical models and score-based generative models have shown promising performance in both sample quality and sample efficiency, which can be combined with numerical SDE solvers or fast ODE solvers to generate samples (Song et al., 2021b; Song & Ermon, 2020; Lu et al., 2022). Score loss are usually optimized by minimizing the Fisher divergence between the gradients of log-density functions (i.e., scores) and the ground truth scores from data, without handling the intractable partition functions (Song et al., 2020; Luo, 2022). This idea serves as the primary inspiration for our score PDE loss.

## 3 FOKKER–PLANCK NEURAL NETWORKS

Considering that optimization, data, and model form the cores of deep learning approaches, we introduce the FPNN framework through these three components. First, we define the score-based Fokker–Planck loss as the objective function to solve SFP equations. Then, steady-state data from

---

[1]We mainly focus on the SFP equation, where the stochastic system reaches an invariant equilibrium distribution and parameters $\boldsymbol{\mu}$, $\boldsymbol{\sigma}$ are independent of time $t$. Examples of PDEs, experimental settings, as well as expressions for $\boldsymbol{\mu}(\boldsymbol{x})$ and $\boldsymbol{\sigma}(\boldsymbol{x})$ can be found in Sec. C.

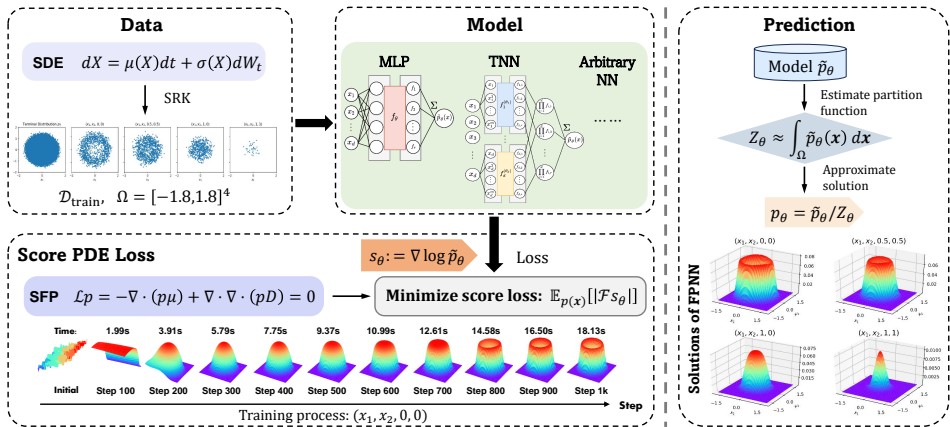

Figure 2: FPNN framework for high-dimensional SFP equations. We use the SRK method to simulate SDE for generating training dataset $\mathcal{D}_{\text{train}}$ and domain $\Omega$. Then, the data is fed into a free-form architecture to produce the score. Next, we train the model by minimizing the score-based FP loss. With the trained model $\widetilde{p}_\theta$, we compute the partition function and perform normalization once to obtain the solution of SFP equation.

true distribution is required to evaluate the score loss, for which we simulate SDEs using the SRK method. Finally, we briefly review the tensor neural network (TNN) and the multi-layer perceptron (MLP), implementing appropriate modifications to suit our task.

## 3.1 SCORE-BASED PDE LOSS

Score matching is originally designed for estimating unnormalized probability densities in machine learning, statistics, and signal processing. The score of density $p(\boldsymbol{x})$ is defined as $\nabla \log p(\boldsymbol{x})$. When the model learns a density $\widetilde{p}_\theta(\boldsymbol{x})$ with the partition function $Z_\theta$, the approximate solution of SFP equation is a normalized density denoted by,

$$p_\theta(\boldsymbol{x}) = \frac{\widetilde{p}_\theta(\boldsymbol{x})}{Z_\theta}, \qquad Z_\theta = \int \widetilde{p}_\theta(\boldsymbol{x}) d\boldsymbol{x} \tag{4}$$

Note that since $\log p_\theta(\boldsymbol{x}) = \log \widetilde{p}_\theta(\boldsymbol{x}) - \log Z_\theta$ and $\nabla \log p_\theta(\boldsymbol{x}) = \nabla \log \widetilde{p}_\theta(\boldsymbol{x})$, we immediately conclude that $p_\theta(\boldsymbol{x})$ and $\widetilde{p}_\theta(\boldsymbol{x})$ share the same score, which does not depend on the intractable partition function $Z_\theta$.

Inspired by this fact, we directly model the unnormalized density $\widetilde{p}_\theta$ and the score function $\boldsymbol{s}_\theta = \nabla \log \widetilde{p}_\theta$ using a neural network, and employ $\boldsymbol{s}_\theta$ to approximate the true score. Once we reformulate the plain PDE loss of SFP equations into a score-based form, our model can learn the correct score function and focus on the shape of PDF without handling NC. With the trained model $\widetilde{p}_\theta(\boldsymbol{x})$, we compute the partition function only once by post-processing and obtain the approximate solution $p_\theta(\boldsymbol{x})$ via Eq. (4).

**Theorem 1** (Score-based FP loss). *Assume the approximate solution $p_\theta(\boldsymbol{x})$ is differentiable and positive, satisfying regularity conditions[2]: for a fixed set of $(\boldsymbol{\mu}(\boldsymbol{x}), \boldsymbol{D}(\boldsymbol{x}))$, $\mathbb{E}[|\mathcal{L}p_\theta(\boldsymbol{x})|]$ is finite for any $\theta$. Denote the model score function as $\boldsymbol{s}_\theta(\boldsymbol{x}) := \nabla \log \widetilde{p}_\theta(\boldsymbol{x})$ to approximate $\nabla \log p(\boldsymbol{x})$ and $p(\boldsymbol{x})$ is the true solution of SFP equation. The plain PDE loss can be expressed as the following up to a constant factor*

$$\mathbb{E}_{p(\boldsymbol{x})}[|\mathcal{F}\boldsymbol{s}_\theta(\boldsymbol{x})|] := \mathbb{E}_{\boldsymbol{x} \sim p(\boldsymbol{x})}[|\boldsymbol{s}_\theta(\boldsymbol{x}) \cdot \widetilde{\boldsymbol{\mu}}(\boldsymbol{x}) + \nabla \cdot \widetilde{\boldsymbol{\mu}}(\boldsymbol{x})|] \tag{5}$$

*where $\widetilde{\boldsymbol{\mu}}(\boldsymbol{x}) := \boldsymbol{\mu}(\boldsymbol{x}) - \nabla \cdot \boldsymbol{D}(\boldsymbol{x}) - \boldsymbol{D}(\boldsymbol{x})\boldsymbol{s}_\theta(\boldsymbol{x})$ does not explicitly involve the density. $\mathcal{F}$ is named as the score-based FP operator, and the proposed PDE loss solely depends on the score $\boldsymbol{s}_\theta(\boldsymbol{x})$.*

---

[2]Since solving FP equations with PINN is a well-posed problem, this result can be directly inferred from the loss function setup. Additionally, the loss functions $\mathbb{E}[|\mathcal{L}p_\theta(\boldsymbol{x})|]$ and $\mathbb{E}_{p(\boldsymbol{x})}[|\mathcal{F}\boldsymbol{s}_\theta(\boldsymbol{x})|]$ share the same minimum value (zero) and uniqueness conditions for the solution under NC.

Proof in Sec. A and we further clarify the connection between the score PDE loss and score matching. By modeling the unnormalized density, we avoid the computational burden caused by frequently estimating $Z_\theta$ at each iteration, and also eliminate the interference of NC during training process. In this way, FPNN decouples the fitting and normalizing stages, ensuring continuous and efficient training process.

## 3.2 STEADY-STATE DATA GENERATION

To evaluate the score loss, we need training data from the true distribution $p(\boldsymbol{x})$ of SFP equation, which can be sampled through the SDE simulation.

**Dataset generation.** Since the SDE is described by a steady-state equation, the stochastic system always reaches an invariant equilibrium distribution under the combined effects of drift and diffusion. Thus, we can freely choose the initial distribution $p_0(\boldsymbol{x})$ and sample an initial point set. These particles then evolve according to the SDE dynamics over a long time $T^3$ to produce a training dataset $\mathcal{D}_{\text{train}}$ from the target distribution $p \approx p_T$.

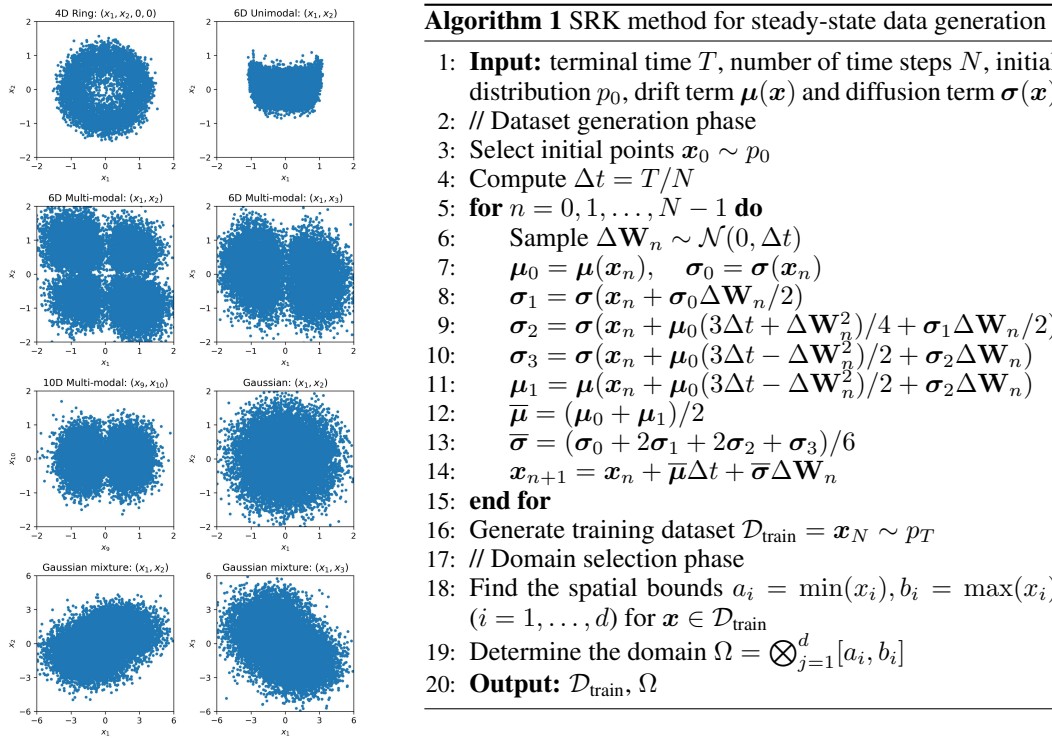

**Algorithm 1** SRK method for steady-state data generation

1: **Input:** terminal time $T$, number of time steps $N$, initial distribution $p_0$, drift term $\boldsymbol{\mu}(\boldsymbol{x})$ and diffusion term $\boldsymbol{\sigma}(\boldsymbol{x})$
2: // Dataset generation phase
3: Select initial points $\boldsymbol{x}_0 \sim p_0$
4: Compute $\Delta t = T/N$
5: **for** $n = 0, 1, \ldots, N - 1$ **do**
6:     Sample $\Delta \mathbf{W}_n \sim \mathcal{N}(0, \Delta t)$
7:     $\boldsymbol{\mu}_0 = \boldsymbol{\mu}(\boldsymbol{x}_n), \quad \boldsymbol{\sigma}_0 = \boldsymbol{\sigma}(\boldsymbol{x}_n)$
8:     $\boldsymbol{\sigma}_1 = \boldsymbol{\sigma}(\boldsymbol{x}_n + \boldsymbol{\sigma}_0 \Delta \mathbf{W}_n / 2)$
9:     $\boldsymbol{\sigma}_2 = \boldsymbol{\sigma}(\boldsymbol{x}_n + \boldsymbol{\mu}_0(3\Delta t + \Delta \mathbf{W}_n^2)/4 + \boldsymbol{\sigma}_1 \Delta \mathbf{W}_n / 2)$
10:     $\boldsymbol{\sigma}_3 = \boldsymbol{\sigma}(\boldsymbol{x}_n + \boldsymbol{\mu}_0(3\Delta t - \Delta \mathbf{W}_n^2)/2 + \boldsymbol{\sigma}_2 \Delta \mathbf{W}_n)$
11:     $\boldsymbol{\mu}_1 = \boldsymbol{\mu}(\boldsymbol{x}_n + \boldsymbol{\mu}_0(3\Delta t - \Delta \mathbf{W}_n^2)/2 + \boldsymbol{\sigma}_2 \Delta \mathbf{W}_n)$
12:     $\overline{\boldsymbol{\mu}} = (\boldsymbol{\mu}_0 + \boldsymbol{\mu}_1)/2$
13:     $\overline{\boldsymbol{\sigma}} = (\boldsymbol{\sigma}_0 + 2\boldsymbol{\sigma}_1 + 2\boldsymbol{\sigma}_2 + \boldsymbol{\sigma}_3)/6$
14:     $\boldsymbol{x}_{n+1} = \boldsymbol{x}_n + \overline{\boldsymbol{\mu}} \Delta t + \overline{\boldsymbol{\sigma}} \Delta \mathbf{W}_n$
15: **end for**
16: Generate training dataset $\mathcal{D}_{\text{train}} = \boldsymbol{x}_N \sim p_T$
17: // Domain selection phase
18: Find the spatial bounds $a_i = \min(x_i), b_i = \max(x_i)$ $(i = 1, \ldots, d)$ for $\boldsymbol{x} \in \mathcal{D}_{\text{train}}$
19: Determine the domain $\Omega = \bigotimes_{j=1}^d [a_i, b_i]$
20: **Output:** $\mathcal{D}_{\text{train}}, \Omega$

Figure 3: *Left*: Cross-sectional views of the training data distributions for SFP equations with complex patterns. *Right*: The SRK method consisting of two phases: dataset generation and domain selection.

**Domain selection.** Based on these samples, we determine an appropriate domain $\Omega$ for SFP problems, as the regions not covered by dataset $\mathcal{D}_{\text{train}}$ have very low or near-zero probability density. This implies that the integral of the true density function outside $\Omega$ (i.e., $1 - \int_\Omega p(\boldsymbol{x})d\boldsymbol{x}$) is approximately zero. And this fact provides a theoretical guarantee for calculating the partition function $Z_\theta$ over $\Omega$. Additionally, training FPNN on $\Omega$ conserves computational resources and avoids the undefined value of $\log \widetilde{p}_\theta(\boldsymbol{x})$ at $\widetilde{p}_\theta(\boldsymbol{x}) = 0$. There is no need to add gated functions or enforce BCs, because the correct score model will ensure that $p_\theta$ is close to 0 at $\partial \Omega$.

We use SRK methods for the strong approximation of SDE, and generate trajectories and samples with the strong order 1.5 Newton scheme (Newton, 1991). The training dataset $\mathcal{D}_{\text{train}}$ and problem domain $\Omega$ are adaptively provided by Algorithm 1. Other numerical approximations for SDE simu-

---

[3]Specifically, $\int_0^T |\boldsymbol{\mu}(\boldsymbol{x}(t))| dt$ needs to be sufficiently large to ensure that the particles thoroughly explore the spatial space and eventually reach a steady state.

lation are alternatives, such as the Euler-Maruyama method and the Milstein method (Bayram et al., 2018). For FPNN, we do not require the data points to exactly follow the true distribution $p(\boldsymbol{x})$, and our loss function offers a degree of tolerance and flexibility. The SRK method and its generated data for different SFP equations are given in Figure 3.

## 3.3 NETWORK ARCHITECTURE

FPNN separates the score model learning from NC, allowing for unrestricted network architectures. In our implementation, we focus on two general approximators: TNN and MLP to model the unnormalized density $\widetilde{p}_\theta(\boldsymbol{x})$. The former is considered for its significant reduction in computational complexity when performing high-dimensional numerical integration for $Z_\theta$. The latter is widely used in deep learning tasks and the integral over domain $\Omega$ can be estimated by Monte Carlo sampling. Both networks are modified to better accommodate complex SFP equations.

### 3.3.1 TENSOR NEURAL NETWORKS

Inspired by SPINN (Cho et al., 2024) and TNN (Wang et al., 2022), we parametrize each spatial component using an separated neural network $f_i : \mathbb{R} \to \mathbb{R}^r$, which takes the coordinates of $i$-th axis as input and produces a $r$-dimensional feature representation. We propose the power embedding (PE) to enhance the representation and better capture high-order drift dynamics. In each dimension, the input $x_i$ is transformed into $[x_i, x_i^2, \cdots, x_i^m]$ before being fed into the network $f_i$ ($m$ is a manually specified hyper-parameter). This embedding enables the model to learn complex PDFs more efficiently and accurately (see Tables 4). The final prediction is obtained by taking an element-wise product of these $r$-dimensional features and summing the result:

$$\widetilde{p}_\theta(\boldsymbol{x}) = \sum_{j=1}^{r} \prod_{i=1}^{d} f_{i,j}(x_i; \theta_i) \tag{6}$$

We employ the activation function $tanh$ for all hidden layers and add an additional $softplus$ activation function before the output layer of the sub-networks $f_i$ to make density $\widetilde{p}_\theta(\boldsymbol{x})$ strictly positive. A $k$-dimensional TNN can be built with $k$ one-dimensional TNNs and approximate any $k$-dimensional vector functions $u : \mathbb{R}^d \to \mathbb{R}^k$. For details on the approximation properties, refer to Theorem 2.1 of TNN (Wang et al., 2022) and Appendix D.4 of SPINN (Cho et al., 2024).

Due to the low-rank structure of density representation in TNN, an efficient and accurate quadrature scheme can be designed for high-dimensional integration. Theorem 2 decomposes the high-dimensional integral of density $\widetilde{p}_\theta(\boldsymbol{x})$ into a series of one-dimensional integrals, and we efficiently compute it using the piece-wise Gauss-Legendre quadrature rule within 10 subintervals.

**Theorem 2** (Partition function calculation). *Given the density $\widetilde{p}_\theta(\boldsymbol{x})$ parameterized by TNN, the partition function $Z_\theta$ is computed as,*

$$Z_\theta \approx \sum_{j=1}^{r} \prod_{i=1}^{d} \left( \sum_{n_i=1}^{N_i} w_i^{(n_i)} f_{i,j}\left(x_i^{(n_i)}; \theta_i\right) \right) \tag{7}$$

*where the nodes $\{x_i^{(n_i)}\}_{n_i=1}^{N_i}$ and weights $\{w_i^{(n_i)}\}_{n_i=1}^{N_i}$ are used for numerical integration in the $i$-th dimension ($i = 1, \ldots, d$).*

### 3.3.2 MULTI-LAYER PERCEPTRON

To illustrate the applicability of our score PDE loss to different network architectures, we parameterize the $d$-dimensional spatial variables using an MLP $f_\theta : \mathbb{R}^d \to \mathbb{R}^r$. The $tanh$ function is used in hidden layers and the softplus function is employed to ensure non-negativity in the output layer. The $r$-dimensional features are summed up to produce the final output:

$$\widetilde{p}_\theta(\boldsymbol{x}) = \sum_{j=1}^{r} f_j(\boldsymbol{x}; \theta) \tag{8}$$

In our experiments, we find that hidden neurons with small activation values are overshadowed and inadequately trained in the original setup. And the modified MLP demonstrates improved learning performance.

Efficient numerical integration in Eq. (7) essentially forms high-dimensional grids using tensor products, which imposes certain requirements on the network architecture. More generally, we provide the Monte Carlo estimation of partition function:

$$Z_\theta \approx \int_\Omega \widetilde{p}_\theta(\boldsymbol{x}) d\boldsymbol{x} = \mathbb{E}_{q(\boldsymbol{x})} \left[ \frac{\widetilde{p}_\theta(\boldsymbol{x})}{q(\boldsymbol{x})} \right] \tag{9}$$

For simplicity, we set $q(\boldsymbol{x})$ as an uniform distribution on $\Omega$, i.e., $q(\boldsymbol{x}) = \frac{1}{\nu(\Omega)}$ for $\boldsymbol{x} \in \Omega$, where $\nu$ denotes the Lebesgue measure. Since the optimization process does not involve normalization operations, we are free to choose the quadrature scheme to compute $Z_\theta$ by post-processing.

## 4 EXPERIMENT

Consider the challenges of solving high-dimensional SFP equations, we evaluate on several benchmark problems with analytical solutions. The 4D Ring is tested in FP solver (Zhai et al., 2022), while the 6D problems and 10D Multi-modal problem are adapted from TFFN (Wang et al., 2024). We further construct the 10D Gaussian mixture distribution (Tang et al., 2022) and 20D Gaussian function to comprehensively test the applicability of our FPNN. Our PDE examples span various 4-20 dimensional steady-state solutions, including ring-shape density, arbitrary potential function, and Gaussian mixture distribution, with complicated interactions among spatial coordinates.

Notably, few recent works can effectively solve such high-dimensional, challenging and different types of SFP equations, without the limitations in Table 1. We utilize TFNN as the baseline for comparison. The models are implemented in PyTorch framework and trained on NVIDIA Quadro RTX 8000 GPU with 48GB memory. Codes are provided in the supplementary material. We set a consistent seed across all experiments to ensure the reproducibility of our results.

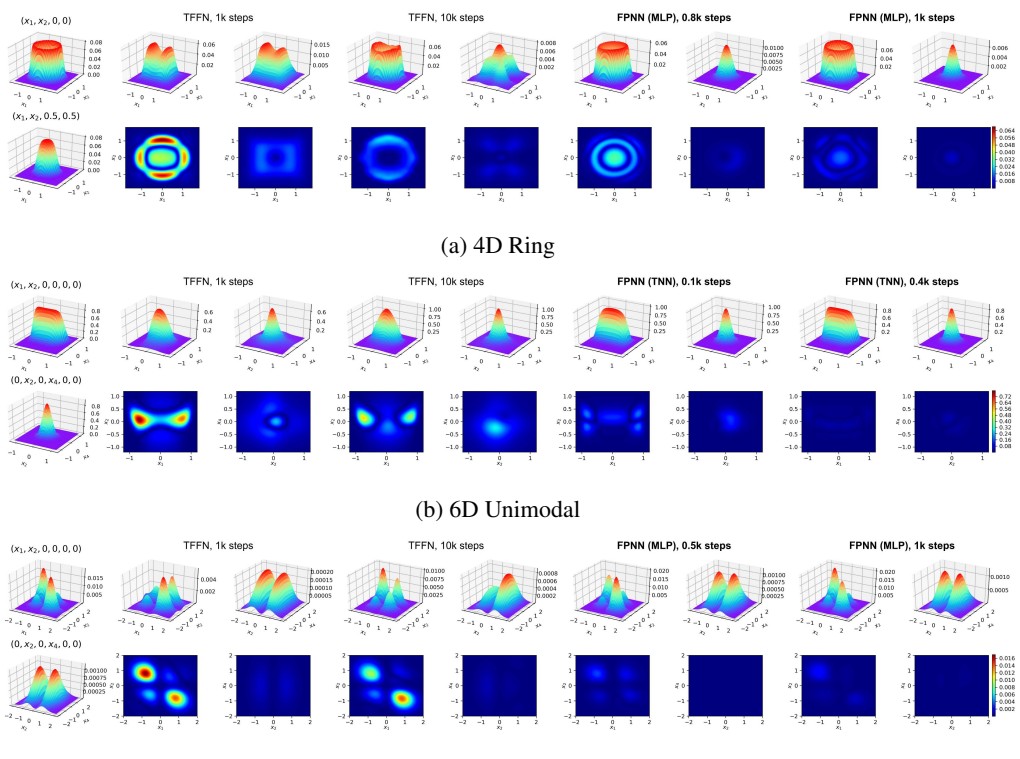

Figure 4: Comparison of efficiency between TFFN and FPNN. The first column displays the true solution, while columns 2-5 and columns 6-9 show the predicted solutions (*top*) and corresponding absolute errors (*bottom*) for TFFN and FPNN during training. Both models utilize the same error colorbar for fair comparison.

## 4.1 Evaluation Metrics

Due to the small solutions to high-dimensional problems, mean absolute error (MAE) is insufficient to accurately evaluate the performance of models in different SFP problems. Even if the network learns a zero solution, MAE is not noticeable and merely equal the average true density on the test dataset, making it difficult to identify this error. However, mean absolute percentage error (MAPE) results in 100%, giving us a great indicator of performance. We incorporate MAPE to measure the consistency between the predicted and true solutions.

For high-dimensional problems, we are limited to visualizing the results using selected slices. However, only testing errors on cross-sectional data is insufficient to measure the performance of high-dimensional solutions due to their multi-modal complexity. Thus, we generate a test dataset $\mathcal{D}_{\text{test}} = \{p(\boldsymbol{x}) > \epsilon, \boldsymbol{x} \in \mathbb{R}^d\}$ to globally evaluate error metrics. To avoid the ineffective MAPE, $\mathcal{D}_{\text{test}}$ is generated by the gradient ascent method on analytical solutions, with a threshold $\epsilon$ to reject extremely small probability densities. This approach is more efficient than the traditional method of randomly sampling spatial points and filtering out those with densities below $\epsilon$. The same test dataset and evaluation metrics are used for FPNN and TFFN, ensuring a fair comparison.

Table 2: Experimental results of TFFN and FPNN on 4-6 dimensional SFP equations.

| SFP equations | Domain $\Omega$ | TFFN | | FPNN (Ours) | |
|---|---|---|---|---|---|
| | | MAE | MAPE | MAE | MAPE |
| 4D Ring | $[-1.8, 1.8]^4$ | $3.61 \times 10^{-3}$ | 49.25% | $5.56 \times 10^{-4}$ | 3.84% |
| 6D Unimodal | $[-1.2, 1.2]^6$ | $4.00 \times 10^{-2}$ | 293% | $1.48 \times 10^{-3}$ | 4.33% |
| 6D Multi-modal | $[-2, 2]^6$ | $1.84 \times 10^{-3}$ | 92.90% | $1.98 \times 10^{-4}$ | 12.18% |

## 4.2 Experimental Analysis

**Score PDE loss.** Without the interference from NC, score loss significantly improves the optimization dynamics, enabling FPNN to quickly learn the correct scores and successfully capture the shape of PDF. This is crucial for the post-process of calculating normalizing constant and is a key factor in the success of our framework. In Figure 5, we plot the training loss $\mathcal{J}_{\text{score}}$ of FPNN for various SFP equations. Unlike the vanilla PINN, the magnitude of $\mathcal{J}_{\text{score}}$ hardly change with increasing dimensionality or decreasing solutions. The score PDE loss consistently remains within a stable range of $10^2$ to $10^{-1}$, demonstrating numerical stability across dimensions. Empirical evidence shows that when $\mathcal{J}_{\text{score}}$ falls below 1, FPNN generally learns the density function well. Thus, score loss also serves as an indicator of training progress and guides the network training regardless of dimensionality.

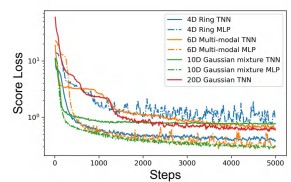

Figure 5: Plot of score PDE loss for 4-20 dimensional SFP equations during training.

Noting that both FPNN and TFFN strictly satisfy NC, we compare two PDE losses $\mathcal{J}_{\text{score}}$ and $\mathcal{J}_{\text{plain}}$ for 4D Ring problem in Figure 6. To balance the numerical scale of loss functions, we evaluate the plain PDE loss for FPNN and TFFN using $10k$ samples. These points are uniformly sampled from $\Omega$ and not used for training in both models. FPNN calculates $Z_\theta$ for normalization based on the current model at each step. Under the same evaluation, we observe that $\mathcal{J}_{\text{plain}}$ of FPNN drops faster and lower, indicating that our model

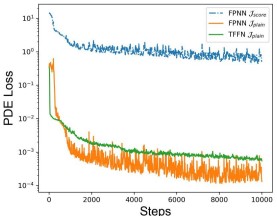

Figure 6: Plot of score PDE loss and plain PDE loss for 4D Ring.

trains more efficiently and stably using the score PDE loss. Moreover, although $\mathcal{J}_{\text{plain}}$ of TFFN also decreases to small values, the shape of approximate solution does not match the true density in Figure 4a. This deviation grows larger in higher-dimensional problems, leading to the failure or inapplicability of existing methods such as PINN and TFFN in high-dimensional SFP problems.

**Network Architecture.** Increasing network capacity and width enhances model performance, resulting in fewer training steps and more accurate solutions. In Table 4 and Table 5, PE shows great

improvements in learning drift functions with high-order polynomial forms. Surprisingly, MLP-based FPNN utilizes 256 parameters to handle the complex 4D Ring problem and MAPE is reduced to 11.36% in only 4.8 minutes. In contrast, TFFN with 33,792 parameters is trained for 27.6 minutes and still fails to achieve comparable performance, as illustrated in Figure 4a. Given the flexibility in model, ResNet (He et al., 2016) is also alternative for deeper networks. In Figure 8, we observe that both TNN-based and MLP-based FPNN correctly capture the two peaks along the $x_9$ axis in 10D Multi-modal problem, requiring only 200 and $1k$ steps, respectively.

Table 3: Comparison of network structure for FPNN on high-dimensional SFP equations.

| SFP equations | Domain $\Omega$ | Model | Parameters | MAE | MAPE |
|---|---|---|---|---|---|
| 10D Multi-modal | $[-1.2, 1.2]^{10}$ | TNN | 84,480 | $1.26 \times 10^{-4}$ | 32.00% |
| | | MLP | 13,184 | $6.53 \times 10^{-5}$ | 18.30% |
| 10D Gaussian mixture | $[-5, 5]^{10}$ | TNN | 126,080 | $1.84 \times 10^{-7}$ | 24.50% |
| | | MLP | 1,480 | $1.16 \times 10^{-7}$ | 18.38% |
| 20D Gaussian | $[-2, 2]^{20}$ | TNN | 252,160 | $1.22 \times 10^{-8}$ | 12.79% |

**Partition Function.** For MLP-based FPNN, we employ Monte Carlo sampling to draw a dataset $\mathcal{D}_Z$ from the distribution $q(\boldsymbol{x})$ and estimate the partition function $Z_\theta$ in Eq.(9). The size of $\mathcal{D}_Z$, denoted by $|\mathcal{D}_Z|$, determines the accuracy of unbiased estimation of $Z_\theta$, which improves as $|\mathcal{D}_Z|$ increases. Figure 7a illustrates how the number of MC samples affects the estimation of $Z_\theta$, and subsequently, the prediction error of our solution.

Due to the randomness, errors fluctuates significantly with small dataset $\mathcal{D}_Z$, especially in the range $|\mathcal{D}_Z| = 5k \sim 20k$. As $|\mathcal{D}_Z|$ increases, the prediction stabilizes. Higher-dimensional densities require more samples to accurately estimate the partition function. For all SFP problems, we set $|\mathcal{D}_Z| = 20k$, which our experiments show is sufficient for the accuracy of FPNN. However, for 10D Gaussian mixture problem, the expanded domain resulted in a MAPE of 58.66% with this setting. After increasing $|\mathcal{D}_Z|$ to $100k$, we get better solution with a MAPE of 18.38%.

For TNN-based FPNN, we also compare the Gauss-Legendre quadrature scheme in Eq.(7) with the MC sampling method in Eq.(9). As shown in Figure 7b, with more samples in $\mathcal{D}_Z$, the MC estimation $Z_{\text{MC}}$ gradually approaches the numerical integration $Z_{\text{GL}}$ (the rightmost column), and MAPE steadily decreases. More comparisons and limitations are provided in Sec. B.

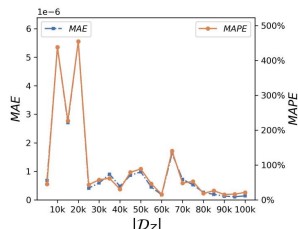

(a) MLP-based FPNN

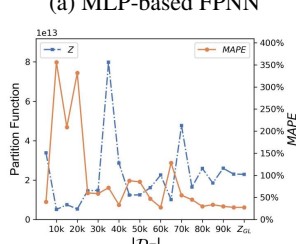

(b) TNN-based FPNN

Figure 7: 10D Gaussian mixture: MAE, MAPE and $Z_\theta$ at different $|\mathcal{D}_Z|$ for FPNN.

| Model | Layers | Parameters | MAE | MAPE |
|---|---|---|---|---|
| TNN | $[m$, hidden layers, $r]$ | | | |
| | $[1, 64, 128]$ | 33,792 | $7.27 \times 10^{-3}$ | 99.82% |
| | $[3, 20, 20, 20, 20]$ | 5,360 | $4.25 \times 10^{-3}$ | 65.41% |
| | $[3, 64, 64, 64]$ | 34,304 | $6.91 \times 10^{-4}$ | 7.58% |
| | $[5, 64, 128]$ | 34,816 | $8.20 \times 10^{-4}$ | 5.66% |
| | $[8, 64, 128]$ | 35,584 | $5.56 \times 10^{-4}$ | 3.84% |
| MLP | $[d$, hidden layers$]$ | | | |
| | $[4, 8, 8, 8]$ | 256 | $1.26 \times 10^{-3}$ | 11.36% |
| | $[4, 20, 20, 20, 20]$ | 1,360 | $9.74 \times 10^{-4}$ | 8.74% |
| | $[4, 64, 64, 64]$ | 8,640 | $8.81 \times 10^{-4}$ | 6.48% |
| | $[4, 128, 128]$ | 17,152 | $6.21 \times 10^{-4}$ | 5.39% |

Table 4: 4D Ring: comparison of FPNN with different networks ($10k$ steps).

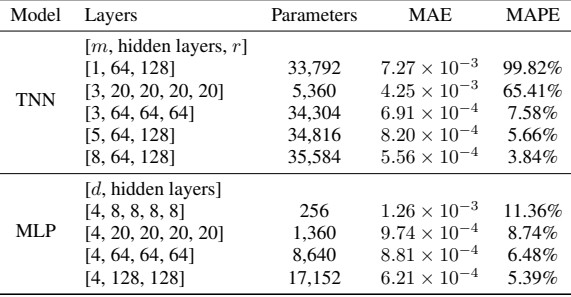



Figure 8: 10D Multi-modal: predicted solution of FPNN during training.

| Model | Layers | Parameters | Steps | MAE | MAPE |
|---|---|---|---|---|---|
| | $[m$, hidden layers, $r]$ | | | | |
| | [1, 20, 20, 20, 20] | 7,800 | $10k$ | $1.26 \times 10^{-2}$ | 30.60% |
| | [1, 64, 64, 64] | 50,688 | $2k$ | $1.33 \times 10^{-2}$ | 31.84% |
| TNN | [3, 64, 64, 64] | 51,456 | $2k$ | $2.94 \times 10^{-3}$ | 9.49% |
| | [5, 64, 64, 64] | 52,224 | $2k$ | $2.44 \times 10^{-3}$ | 8.00% |
| | [8, 64, 128] | 53,376 | $2k$ | $2.18 \times 10^{-3}$ | 7.03% |
| | [8, 64, 128] | 53,376 | $10k$ | $1.48 \times 10^{-3}$ | 4.33% |
| | $[d$, hidden layers] | | | | |
| MLP | [6, 8, 8, 8] | 200 | $4k$ | $2.46 \times 10^{-2}$ | 41.64% |
| | [6, 20, 20, 20] | 980 | $6k$ | $5.40 \times 10^{-3}$ | 13.87% |
| | [6, 64, 64, 64] | 8,768 | $10k$ | $2.94 \times 10^{-3}$ | 8.61% |

Table 5: 6D Unimodal: comparison of FPNN with different networks.

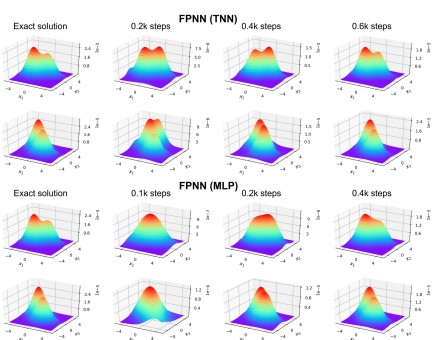

Figure 9: 10D Gaussian mixture: predicted solution of FPNN during training.

**Computational Efficiency.** Our computational complexity is significantly reduced compared to existing methods, because the normalization condition does not need to be explicitly considered during training. For PDE residuals, we only perform an additional logarithmic operation on the output layer, and the computational costs for $\mathcal{J}_{\text{plain}}$, $\mathcal{J}_{\text{score}}$ and their computational graphs remain roughly comparable. With the improved optimization dynamics and post-process of normalization, FPNN reduces both the total number of training epochs and the computational cost per iteration, achieving faster efficiency and enhanced performance.

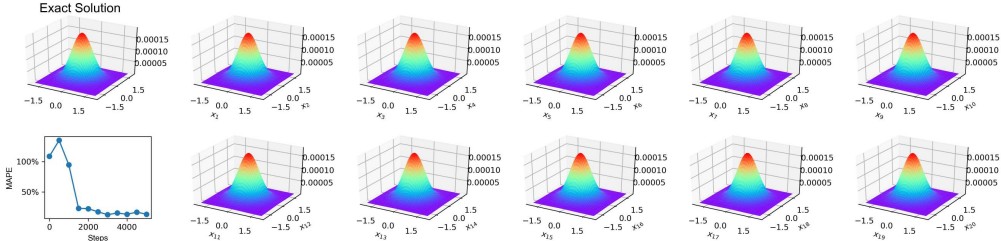

Figure 10: 20D Gaussian: predicted solution of FPNN. The first column shows the exact solution and the changes of MAPE during training. Due to symmetry, the cross-sections of any two dimensions near the origin exhibit the same shape. The remaining parts display the predicted densities for pairs of adjacent dimensions, such as $(x_1, x_2, 0, \ldots, 0)$, $(0, 0, x_3, x_4, 0, \ldots, 0)$, and so on.

## 5 LIMITATIONS AND FUTURE WORK

Despite comprehensive experimental results show the effectiveness of our FPNN framework on challenging high-dimensional steady-state FP equations, there remain challenges on solving time-dependent FP equations. Time-varying probability densities are often non-localized and it is difficult to identify a suitable integration domain at any given time and compute the time-varying normalizing function. Our exploration is merely a promising start, and we look forward to its applications in broader fields, such as physics, finance, diffusion models, and mean-field games. In the future, we aim to develop an efficient solver for general Fokker-Planck equations.

## 6 CONCLUSION

We propose a novel framework, FPNN, that decouples the normalization condition through score PDE loss for solving high-dimensional SFP equations. FPNN eliminates the computation and influence of the partition function for more efficient and stable training. In addition, arbitrary network architectures are allowed to strictly enforce normalization constraints, without sacrificing representation power. Our method outperforms existing methods in both training efficiency and prediction accuracy, demonstrating strong potential and general applicability.

ACKNOWLEDGMENTS

This work was supported by the National Science and Technology Major Project under Grant 2022ZD0116401, the National Natural Science Foundation of China under Grant 62141605, and the Fundamental Research Funds for the Central Universities, China.

DATA AVAILABILITY

We provide the code for generating test datasets and fix the random seed for reproducibility, enabling future research to perform comparisons with FPNN under consistent evaluation metrics. Our code is available at https://github.com/niuffs/FPNN.

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

## A    Loss functions for SFP equations

We first introduce the loss functions and notations listed in Table 1 to provide the background for SFP equations. Then, we present the derivation of the score-based FP loss, which constitutes the core innovation and contribution of our work. Furthermore, we explain the relationship between the score PDE loss of SFP equations and the score-matching loss of generative models.

### A.1    Related works

PINN is a general deep-learning framework for solving PDEs and has achieved significant success in various problems, such as the Navier-Stokes equations, Allen-Cahn equation, Schrödinger equation, etc. However, PINN faces challenges in Fokker-Planck equations, with the plain PDE loss:

$$\mathcal{J}_{\text{plain}}(p_\theta) = \mathbb{E}\left[|\nabla \cdot (p_\theta \boldsymbol{\mu}) - \nabla \cdot (\nabla \cdot (D p_\theta))|\right] \tag{10}$$

It is evident that the zero solution $p_\theta = 0$ also satisfies this equation. Therefore, directly minimizing $\mathcal{J}_{\text{plain}}$ often leads to network collapse, where $p_\theta$ converges to a trivial solution. Data-driven methods rely on reference solutions $\{(\boldsymbol{x}_j, p_j)\}_{j=1}^N$ and introduce the regression loss:

$$\mathcal{J}_{\text{label}} = \frac{1}{N} \sum_{j=1}^N (p_\theta(\boldsymbol{x}_j) - p_j)^2 \tag{11}$$

Without labeled data, existing strategies generally fall into two categories involving normalization constraints in soft or hard manner. Soft constraints add a normalization penalty term to enforce $Z_\theta = 1$, thereby enabling the use of Eq.(9). This involves sampling a dataset $\mathcal{D}_{\text{norm}}$ from the distribution $q(\boldsymbol{x})$, with the number of samples given by $|\mathcal{D}_{\text{norm}}|$. In our experiments, $\Omega = \bigotimes_{j=1}^d [a_i, b_i]$ is a hypercube with the volume of $\nu(\Omega) = \prod_{i=1}^d |b_i - a_i|$. The normalization loss is defined as:

$$\mathcal{J}_{\text{norm}} = \left(\frac{\nu(\Omega)}{|\mathcal{D}_{\text{norm}}|} \sum_{\boldsymbol{x} \in \mathcal{D}_{\text{norm}}} p_\theta(\boldsymbol{x}) - 1\right)^2 \tag{12}$$

Hard constraints, on the other hand, develop specific structures to represent the density function, such as normalizing flows and their variants. The flow-based generative modeling is to seek an invertible mapping $\boldsymbol{z} = f(\boldsymbol{x})$ and the PDF of $\boldsymbol{x}$ follows the change of variables formula to strictly adhere NC.

$$p_X(\boldsymbol{x}) = p_Z(\boldsymbol{z})|\det \nabla_{\boldsymbol{x}} f(\boldsymbol{x})| \tag{13}$$

However, the former method requires manual balancing of multi-objective losses, while the latter may constrain the model's representation capacity. Furthermore, in terms of computational efficiency, TFFN requires estimating $Z_\theta$ at every iteration, "soft" PINN necessitates computing $\mathcal{J}_{\text{norm}}$ at each step, and density estimation with normalizing flows involves simultaneous tracking the changes of both $x$ and log-density.

FPNN deviates from all existing approaches by changing the loss function directly. Our score PDE loss allows to model the unnormalized density $\widetilde{p}_\theta$ and decouples NC from the training process. Specifically, we replace $\mathcal{J}_{\text{plain}}$ with the following score PDE loss:

$$\mathcal{J}_{\text{score}}(\boldsymbol{s}_\theta) = \mathbb{E}\left[|\nabla \log \widetilde{p}_\theta \cdot \widetilde{\boldsymbol{\mu}}(\boldsymbol{x}) + \nabla \cdot \widetilde{\boldsymbol{\mu}}(\boldsymbol{x})|\right],$$
$$\widetilde{\boldsymbol{\mu}}(\boldsymbol{x}) = \boldsymbol{\mu}(\boldsymbol{x}) - \nabla \cdot D(\boldsymbol{x}) - D(\boldsymbol{x})\nabla \log \widetilde{p}_\theta \tag{14}$$

In this form, the trivial solution $\widetilde{p}_\theta = 0$ no longer satisfies the equation. The score PDE loss maintains equivalence with the plain PDE loss while avoiding a zero solution and circumventing $\mathcal{J}_{\text{label}}, \mathcal{J}_{\text{norm}}$, or the normalization condition in loss function which causes difficulty in training.

### A.2    Derivation of score PDE loss

**Theorem 1** (Score-based FP loss). *Assume the approximate solution $p_\theta(\boldsymbol{x})$ is differentiable and positive, satisfying regularity conditions: for a fixed set of $(\boldsymbol{\mu}(\boldsymbol{x}), D(\boldsymbol{x}))$, $\mathbb{E}[|\mathcal{L}p_\theta(\boldsymbol{x})|]$ is finite for any $\theta$. Denote the model score function as $\boldsymbol{s}_\theta(\boldsymbol{x}) := \nabla \log \widetilde{p}_\theta(\boldsymbol{x})$ to approximate $\nabla \log p(\boldsymbol{x})$ and*

*$p(\boldsymbol{x})$ is the true solution of SFP equation. The plain PDE loss can be expressed as the following up to a constant factor*

$$\mathbb{E}_{p(\boldsymbol{x})}[|\mathcal{F}\boldsymbol{s}_\theta(\boldsymbol{x})|] := \mathbb{E}_{\boldsymbol{x}\sim p(\boldsymbol{x})}[|\boldsymbol{s}_\theta(\boldsymbol{x})\cdot\widetilde{\boldsymbol{\mu}}(\boldsymbol{x}) + \nabla\cdot\widetilde{\boldsymbol{\mu}}(\boldsymbol{x})|] \tag{15}$$

*where $\widetilde{\boldsymbol{\mu}}(\boldsymbol{x}) := \boldsymbol{\mu}(\boldsymbol{x}) - \nabla\cdot\boldsymbol{D}(\boldsymbol{x}) - \boldsymbol{D}(\boldsymbol{x})\boldsymbol{s}_\theta(\boldsymbol{x})$ does not explicitly involve the density. $\mathcal{F}$ is named as the score-based FP operator, and the proposed PDE loss solely depends on the score $\boldsymbol{s}_\theta(\boldsymbol{x})$.*

*Proof.* First, for any positive probability density $p(\boldsymbol{x})$, note that $\mathcal{L}p(\boldsymbol{x})$ can be expanded to

$$
\begin{aligned}
\mathcal{L}p(\boldsymbol{x}) &= -\nabla\cdot(p(\boldsymbol{x})\boldsymbol{\mu}(\boldsymbol{x})) + \nabla\cdot[\nabla\cdot(\boldsymbol{D}(\boldsymbol{x})p(\boldsymbol{x}))] \\
&= -\nabla\cdot(p(\boldsymbol{x})\boldsymbol{\mu}(\boldsymbol{x}) - \nabla\cdot(\boldsymbol{D}(\boldsymbol{x})p(\boldsymbol{x}))) \\
&= -\nabla\cdot(p(\boldsymbol{x})\boldsymbol{\mu}(\boldsymbol{x}) - (\nabla\cdot\boldsymbol{D}(\boldsymbol{x}))p(\boldsymbol{x}) - \boldsymbol{D}(\boldsymbol{x})\nabla p(\boldsymbol{x})) \\
&= -\nabla\cdot(p(\boldsymbol{x})(\boldsymbol{\mu}(\boldsymbol{x}) - \nabla\cdot\boldsymbol{D}(\boldsymbol{x}) - \boldsymbol{D}(\boldsymbol{x})\nabla\log p(\boldsymbol{x}))) \\
&= -\nabla\cdot(p(\boldsymbol{x})\widetilde{\boldsymbol{\mu}}(\boldsymbol{x})) \\
&\overset{i.e.}{=} -\sum_{i=1}^d \frac{\partial}{\partial x_i}[p(\boldsymbol{x})\widetilde{\mu}_i(\boldsymbol{x})]
\end{aligned}
\tag{16}
$$

where we define

$$\widetilde{\boldsymbol{\mu}}(\boldsymbol{x}) := \boldsymbol{\mu}(\boldsymbol{x}) - \nabla\cdot\boldsymbol{D}(\boldsymbol{x}) - \boldsymbol{D}(\boldsymbol{x})\nabla\log p(\boldsymbol{x}) \tag{17}$$

For an approximate solution $p_\theta(\boldsymbol{x})$, the plain FP loss measures the expectation of PDE residuals $\mathcal{J}_{\text{plain}} = \mathbb{E}_{q(\boldsymbol{x})}[|\mathcal{L}p_\theta(\boldsymbol{x})|]$, where $q(\boldsymbol{x})$ can be selected as the uniform distribution on $\Omega$. Based on Eq.(16), we can rewrite this loss to obtain

$$
\begin{aligned}
\mathcal{J}_{\text{plain}}(p_\theta) &= \mathbb{E}_{q(\boldsymbol{x})}[|\mathcal{L}p_\theta(\boldsymbol{x})|] = \int_\Omega \frac{1}{\nu(\Omega)}|\mathcal{L}p_\theta(\boldsymbol{x})|d\boldsymbol{x} \\
&= \frac{1}{\nu(\Omega)}\int_\Omega |\nabla p_\theta(\boldsymbol{x})\cdot\widetilde{\boldsymbol{\mu}}(\boldsymbol{x}) + p_\theta(\boldsymbol{x})(\nabla\cdot\widetilde{\boldsymbol{\mu}}(\boldsymbol{x}))|d\boldsymbol{x} \\
&= \frac{1}{\nu(\Omega)}\int_\Omega p_\theta(\boldsymbol{x})|\nabla\log p_\theta(\boldsymbol{x})\cdot\widetilde{\boldsymbol{\mu}}(\boldsymbol{x}) + \nabla\cdot\widetilde{\boldsymbol{\mu}}(\boldsymbol{x})|d\boldsymbol{x} \\
&= \frac{1}{\nu(\Omega)}\mathbb{E}_{p_\theta(\boldsymbol{x})}[|\nabla\log p_\theta(\boldsymbol{x})\cdot\widetilde{\boldsymbol{\mu}}(\boldsymbol{x}) + \nabla\cdot\widetilde{\boldsymbol{\mu}}(\boldsymbol{x})|] \\
&\approx \frac{1}{\nu(\Omega)}\mathbb{E}_{p(\boldsymbol{x})}[|\mathcal{F}\boldsymbol{s}_\theta(\boldsymbol{x})|] \\
&= \frac{1}{\nu(\Omega)}\mathcal{J}_{\text{score}}(\boldsymbol{s}_\theta)
\end{aligned}
\tag{19}
$$

Our goal is to approximate the solution $p(\boldsymbol{x})$ and score $\nabla\log p(\boldsymbol{x})$ using $p_\theta(\boldsymbol{x})$ and $\boldsymbol{s}_\theta(\boldsymbol{x})$. Thus, we replace the expectation of $p_\theta$ with the ground truth $p$, allowing us to train the score model better. For SFP equations, we easily obtain samples from the invariant distribution $p(\boldsymbol{x})$ through SDE simulation. When $p_\theta$ is close to $p$, we observe that, up to a constant factor of $1/\nu(\Omega)$, the score-based FP loss is equivalent to the plain FP loss of PINN. But this form circumvents the need to handle the partition function and normalization condition. □

If we sample training data over a large spatial domain without prior knowledge, regions with extremely low density may fall below machine precision, resulting in $p_\theta = 0$ and "Not a Number" (NaN) errors for $\log p_\theta$. But it has no impact on the prediction stage, as there is no logarithmic operation. By using the SRK method to generate high-probability training data of the steady-state distribution, we define the domain of SFP problems and avoid this issue.

To address numerical issues, another effective approach is to directly model the log-density $\log p$ using a neural network, which can be implemented with the score PDE loss. Assuming that the computer exactly represents positive values in the range $[\epsilon, K]$, this gives the maximum range of density $p$ as the output. When we model $\log p$, the range is $[-K, K]$ and the corresponding density is in $[e^{-K}, e^K]$, which is significantly larger than the previous range. This approach helps to mitigate the occurrence of NaN values during training.

Sometimes, we can enhance training by adding weights $w(\boldsymbol{x})$ according to the probability density, and generalize the objective function as $\mathbb{E}_{p(\boldsymbol{x})}[|\mathcal{F}\boldsymbol{s}_\theta(\boldsymbol{x})|^\alpha], (0 < \alpha \le 1)$. We set $\alpha = 1$ in our experiments, i.e., $\ell_1$ norm for score loss function. Here is the following derivation:

$$
\begin{aligned}
\mathbb{E}_{p(\boldsymbol{x})}[|\mathcal{F}\boldsymbol{s}_\theta(\boldsymbol{x})|^\alpha] &= \int_\Omega p(\boldsymbol{x})|\mathcal{F}\boldsymbol{s}_\theta(\boldsymbol{x})|^\alpha d\boldsymbol{x} \\
&\approx \nu(\Omega) \int_\Omega \frac{1}{\nu(\Omega)} p(\boldsymbol{x})^{1-\alpha}[p_\theta(\boldsymbol{x})^\alpha|\mathcal{F}\boldsymbol{s}_\theta(\boldsymbol{x})|^\alpha]d\boldsymbol{x} \\
&= \nu(\Omega)\mathbb{E}_{q(\boldsymbol{x})}[p(\boldsymbol{x})^{1-\alpha}|\mathcal{L}p_\theta(\boldsymbol{x})|^\alpha] \\
&= \nu(\Omega)\mathbb{E}_{q(\boldsymbol{x})}[w(\boldsymbol{x})|\mathcal{L}p_\theta(\boldsymbol{x})|^\alpha]
\end{aligned}
\tag{20}
$$

### A.3 CONNECTION BETWEEN SCORE PDE LOSS AND SCORE MATCHING

To illustrate this connection, we consider a special case where $D(\boldsymbol{x}) = I_d$. If minimizing the score loss $\mathcal{J}_{\text{score}}$ drives $\widetilde{\boldsymbol{\mu}}(\boldsymbol{x})$ to zero, the training process becomes equivalent to optimizing the objective function:

$$
\mathbb{E}_{p_{\text{data}}}[\|\boldsymbol{\mu}(\boldsymbol{x}) - \nabla \log \widetilde{p}_\theta\|_2^2]
\tag{21}
$$

Thus, FPNN inherently integrates score matching while preserving the original Fokker-Planck equation and corresponding physical laws.

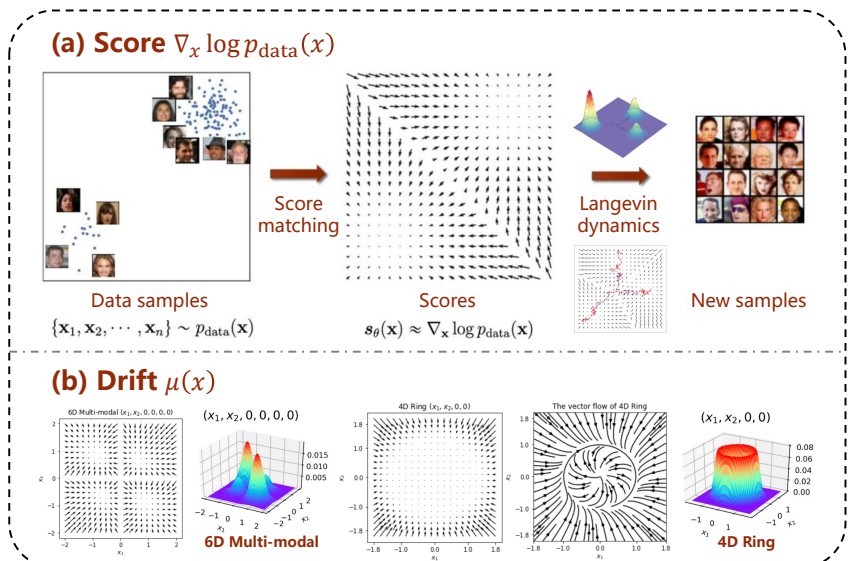

Figure 11: (a) the score in generative models and (b) the drift in SFP equations. Both represent the gradient information of target distributions and indicate regions of high probability.

**Score matching.** In Song's work (Song et al., 2021b), the Noise Conditional Score Network (NCSN) uses a weighted sum of denoising score matching objectives:

$$
\theta^* = \arg\min_\theta \sum_{i=1}^N \sigma_i^2 \mathbb{E}_{p_{\text{data}}(\boldsymbol{x})} \mathbb{E}_{p_{\sigma_i}(\widetilde{\boldsymbol{x}}|\boldsymbol{x})}[\|\boldsymbol{s}_\theta(\widetilde{\boldsymbol{x}}, \sigma_i) - \nabla_{\widetilde{\boldsymbol{x}}} \log p_{\sigma_i}(\widetilde{\boldsymbol{x}}|\boldsymbol{x})\|_2^2]
\tag{22}
$$

and Denoising Diffusion Probabilistic Model (DDPM) leverages a re-weighted variant of the evidence lower bound (ELBO):

$$
\theta^* = \arg\min_\theta \sum_{i=1}^N (1-\alpha_i) \mathbb{E}_{p_{\text{data}}(\boldsymbol{x})} \mathbb{E}_{p_{\alpha_i}(\widetilde{\boldsymbol{x}}|\boldsymbol{x})}[\|\boldsymbol{s}_\theta(\widetilde{\boldsymbol{x}}, i) - \nabla_{\widetilde{\boldsymbol{x}}} \log p_{\alpha_i}(\widetilde{\boldsymbol{x}}|\boldsymbol{x})\|_2^2]
\tag{23}
$$

It can be seen that our equivalent score-matching loss:

$$
\theta^* = \arg\min_\theta \mathbb{E}_{p_{\text{data}}(\boldsymbol{x})}[\|\boldsymbol{s}_\theta(\boldsymbol{x}) - \boldsymbol{\mu}(\boldsymbol{x})\|_2^2]
\tag{24}
$$

shares a similar form with score-based generative models. These generative models gradually learn the score of the perturbed data distribution (i.e., conditional Gaussian distribution), while our FPNN directly learns the known drift $\boldsymbol{\mu}(\boldsymbol{x})$ for solving FP equations.

### A.4 GENERATIVE MODELS AND FPNN

For score-based generative modeling, we consider three distinct dynamical equations:

- **Forward SDE.**
$$dx = \boldsymbol{f}(\boldsymbol{x}, t)dt + \boldsymbol{g}(t)d\mathbf{w} \tag{25}$$

The probability density function $p(\boldsymbol{x}, t)$ satisfies the FP equation:

$$\frac{\partial p}{\partial t} = -\nabla \cdot (\boldsymbol{f}p) + \nabla \cdot \nabla \cdot \left( \frac{1}{2}\boldsymbol{g}\boldsymbol{g}^T p \right) \tag{26}$$

- **Probability flow ODE.**

$$dx = \left( \boldsymbol{f}(\boldsymbol{x}, t) - \frac{1}{2}\boldsymbol{g}(t)\boldsymbol{g}^T(t)\nabla \log p \right) dt \tag{27}$$

Substituting the drift and zero-diffusion term yields the FP equation:

$$\frac{\partial p}{\partial t} = -\nabla \cdot \left[ \left( \boldsymbol{f}(\boldsymbol{x}, t) - \frac{1}{2}\boldsymbol{g}(t)\boldsymbol{g}^T(t)\nabla \log p \right) p \right] \tag{28}$$

$$= -\nabla \cdot (\boldsymbol{f}p) + \nabla \cdot \nabla \cdot \left( \frac{1}{2}\boldsymbol{g}\boldsymbol{g}^T p \right) \tag{29}$$

which is identical to Eq.(25). Here, $\boldsymbol{g}(t)$ is independent of $\boldsymbol{x}$ and it is important to emphasize that $\nabla \log p$ must correspond to the score of density $p(\boldsymbol{x}, t)$ induced by the forward SDE to ensure consistency in simplifying Eq.(28). Since there is no diffusion term, the probability flow (PF) ODE is reversible, allowing for an efficient denoising process.

- **Reverse SDE.**
$$dx = \left( \boldsymbol{f}(\boldsymbol{x}, t) - \boldsymbol{g}(t)\boldsymbol{g}^T(t)\nabla \log p \right) dt + \boldsymbol{g}(t)d\mathbf{w} \tag{30}$$

Compared to the PF ODE, the reverse SDE introduces additional randomness to denoising processes, thereby increasing the diversity of samples. Anderson (1982) provided the detailed derivation. Here, we give a less rigorous but intuitive explanation for clarity. Let the drift term of the PF ODE and the reverse SDE be $\widetilde{f}_1(\boldsymbol{x}, t)$ and $\widetilde{f}_2(\boldsymbol{x}, t)$. We consider a time reversion $t = T - \tau$ and obtain: $dx = -\widetilde{\boldsymbol{f}}_2(\boldsymbol{x}, T-\tau)d\tau + \boldsymbol{g}(T-\tau)d\mathbf{w}$. The corresponding FP equation is:

$$\frac{\partial p}{\partial \tau} = -\nabla \cdot \left( -\widetilde{\boldsymbol{f}}_2 p \right) + \nabla \cdot \nabla \cdot \left( \frac{1}{2}\boldsymbol{g}\boldsymbol{g}^T p \right)$$

$$= \nabla \cdot \left[ (\boldsymbol{f} - \boldsymbol{g}\boldsymbol{g}^T \nabla \log p)p + \left( \frac{1}{2}\boldsymbol{g}\boldsymbol{g}^T \right) \nabla p \right]$$

$$= \nabla \cdot \left( \widetilde{\boldsymbol{f}}_1(\boldsymbol{x}, T-\tau)p \right) \tag{31}$$

When expressed in terms of $t$, this becomes consistent with Eq.(28):

$$\frac{\partial p}{\partial t} = -\nabla \cdot \left( \widetilde{\boldsymbol{f}}_1(\boldsymbol{x}, t)p \right) \tag{32}$$

For generative models, both the drift term $\boldsymbol{f}$ and the diffusion term $\boldsymbol{g}$ are typically predefined. EDM (Karras et al., 2022) replaces the functions $f(t)$ and $g(t)$ with the geometrically meaningful scale schedule $s(t)$ and noise schedule $\sigma(t)$. In summary, the forward SDE is only applicable for progressively adding noise to a dataset and cannot be used for sampling images. To generate high-quality samples, we need to adopt reversible PF ODE or reverse-time SDE, both of which require

pre-estimation of the unknown scores via score matching. When the score of the target distribution is not analytically available, sliced score matching (Song et al., 2020; 2021a) can be used:

$$\mathbb{E}_{p_t(\boldsymbol{x})}[\|\boldsymbol{s}_\theta(\boldsymbol{x}, t) - \nabla \log p_t(\boldsymbol{x})\|^2]$$
$$= \mathbb{E}_{p_t(\boldsymbol{x})}[\|\boldsymbol{s}_\theta(\boldsymbol{x}, t)\|^2 - 2\boldsymbol{s}_\theta(\boldsymbol{x}, t) \cdot \nabla \log p_t(\boldsymbol{x}) + \|\nabla \log p_t(\boldsymbol{x})\|^2] \tag{33}$$
$$= \mathbb{E}_{p_t(\boldsymbol{x})}[\|\boldsymbol{s}_\theta(\boldsymbol{x}, t)\|^2 + 2\nabla \cdot \boldsymbol{s}_\theta(\boldsymbol{x}, t)] + C$$
$$\theta^* = \arg \min_\theta \mathbb{E}_{p_t(\boldsymbol{x})}[\|\boldsymbol{s}_\theta(\boldsymbol{x}, t)\|^2 + 2\nabla \cdot \boldsymbol{s}_\theta(\boldsymbol{x}, t)] \tag{34}$$

The relative abundance of samples contains the gradient information of the log-density in the data space. This approach only uses data samples to estimate the correct score. The ultimate goal of both reverse-time PF ODE and SDE is to map from the prior Gaussian distribution $p_G(\boldsymbol{x})$ to the target data distribution $p_{\text{data}}(\boldsymbol{x})$.

**Flow matching.** Different from score matching, another approach is to learn the vector field $\boldsymbol{\phi}_\theta(\boldsymbol{x}, t) : \mathbb{R}^{d+1} \to \mathbb{R}^d$ that helps transform the prior distribution $p_G$ to the data distribution $p_{\text{data}}$. This method bypasses the unknown $\nabla \log p$ and directly learns the complete drift term $\boldsymbol{\phi}_\theta := \boldsymbol{f} - \frac{1}{2}\boldsymbol{g}\boldsymbol{g}^T\nabla \log p$, resulting in the following equation:

$$\frac{d\boldsymbol{x}}{dt} = \boldsymbol{\phi}_\theta(\boldsymbol{x}, t) \tag{35}$$

We can train the network $\boldsymbol{\phi}_\theta$ in various ways. For instance, continuous normalizing flows (Chen et al., 2018) match the distributions at the endpoint and optimize the network parameters by minimizing the Kullback-Leibler (KL) divergence:

$$\frac{d \log p(\boldsymbol{x}(t), t)}{dt} = -\nabla \cdot \boldsymbol{\phi}_\theta(\boldsymbol{x}, t) \tag{36}$$
$$\theta^* = \arg \min_\theta \text{KL}(p_0^\theta(\boldsymbol{x}) \| p_G(\boldsymbol{x})) \tag{37}$$

where $p_0^\theta(\boldsymbol{x})$ is transformed from $p_{\text{data}}(\boldsymbol{x})$ by the instantaneous change of variables. Alternatively, we can match trajectories and vector fields at intermediate times, such as flow matching (Lipman et al., 2022; Liu et al., 2022):

$$\theta^* = \arg \min_\theta \mathbb{E}_{t \sim U[0,T], \boldsymbol{x} \sim p_t(\boldsymbol{x})}[\|\boldsymbol{u}_t(\boldsymbol{x}) - \boldsymbol{\phi}_\theta(\boldsymbol{x}, t)\|^2] \tag{38}$$

where $\boldsymbol{u}_t$ is the desired transport velocity and usually produces the straight paths.

**Generative models.** We provide a new perspective on the relationship between diffusion models and GANs. If the denoising process of image generation is treated as gradient ascent along $\nabla \log p_t(\boldsymbol{x})$, then score-based **diffusion models** move $\boldsymbol{x}$ to increase $p_t$ while gradually reducing time $t$, and finally generate high-probability samples under the distribution $p_0 := p_{\text{data}}$. For **generative adversarial networks** (GANs) (Goodfellow et al., 2014; Huang et al., 2025), the discriminator directly models the data distribution $p_{\text{data}} := p_0$, and the generator training can be regarded as performing gradient ascent on $p_0$ in the data space to generate high-probability samples.

These methods generate samples from $p_{\text{data}}(\boldsymbol{x})$, but optimization spaces are different, resulting in different paths. Diffusion models uses trajectories induced by $\{p_t(\boldsymbol{x}), t : T \to 0\}$, while GANs are guided by $p_0(\boldsymbol{x})$.

**FPNN.** The SFP equation is related to a known SDE $d\boldsymbol{x} = \boldsymbol{\mu}(\boldsymbol{x})dt + \boldsymbol{\sigma}d\mathbf{w}$. Traditional PINNs directly minimize the PDE residual to train an approximate solution. Our approach reformulates the forward SDE into the PF ODE:

$$d\boldsymbol{x} = \left(\boldsymbol{\mu}(\boldsymbol{x}) - \frac{1}{2}\boldsymbol{\sigma}\boldsymbol{\sigma}^T \nabla \log p(\boldsymbol{x})\right) dt = \widetilde{f}(\boldsymbol{x})dt \tag{39}$$

and introduces the score function associated with the solution of FP equation. Due to the stationary invariant property, we can set $\widetilde{f}(\boldsymbol{x}) = 0$ to train the network and this reduces to Eq.(21) for $\frac{1}{2}\boldsymbol{\sigma}\boldsymbol{D} = \boldsymbol{\sigma}^T = \boldsymbol{I}$.

In the PF ODEs, the score of generative models and the drift of FP equations share a formal consistency. The drift term $\widetilde{f}(\boldsymbol{x}, t)$ is non-zero in the former and drives towards the data distribution, while $\widetilde{f}(\boldsymbol{x}) = 0$ maintains a steady state in the latter. The former is data-driven, whereas the latter is guided by physical laws.

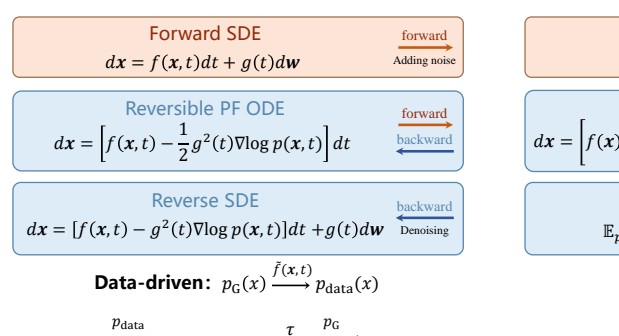

Figure 12: The tasks of image generation and SFP equations.

## B   COMPUTATION OF PARTITION FUNCTION

The main proof is derived from Section 3 of Wang et al. (2022). Here, we demonstrate the simplified integral expression relevant to our work, and the conclusion remains valid for the power embedding.

**Theorem 2** (Partition function calculation). *Given the density $\widetilde{p}_\theta(\boldsymbol{x})$ parameterized by TNN, the partition function $Z_\theta$ is computed as,*

$$Z_\theta \approx \sum_{j=1}^{r} \prod_{i=1}^{d} \left( \sum_{n_i=1}^{N_i} w_i^{(n_i)} f_{i,j} \left( x_i^{(n_i)}; \theta_i \right) \right) \tag{40}$$

*where the nodes $\{x_i^{(n_i)}\}_{n_i=1}^{N_i}$ and weights $\{w_i^{(n_i)}\}_{n_i=1}^{N_i}$ are used for numerical integration in the $i$-th dimension $(i = 1, \ldots, d)$.*

*Proof.* For the unnormalized density $\widetilde{p}_\theta(\boldsymbol{x})$ represented by TNN, the computation of $Z_\theta$ can be decomposed into $d$ one-dimensional integrals:

$$Z_\theta \approx \int_{\Omega} \widetilde{p}_\theta(\boldsymbol{x}) d\boldsymbol{x}$$

$$= \int_{\Omega} \sum_{j=1}^{r} \prod_{i=1}^{d} f_{i,j}(x_i) d\boldsymbol{x}$$

$$= \sum_{j=1}^{r} \int_{\Omega} \prod_{i=1}^{d} f_{i,j}(x_i) d\boldsymbol{x} \tag{41}$$

$$= \sum_{j=1}^{r} \int_{a_d}^{b_d} \cdots \int_{a_1}^{b_1} f_{1,j}(x_1) \cdots f_{d,j}(x_d) dx_1 \cdots dx_d$$

$$= \sum_{j=1}^{r} \left( \int_{a_1}^{b_1} f_{1,j}(x_1) dx_1 \right) \cdots \left( \int_{a_d}^{b_d} f_{d,j}(x_d) dx_d \right) \tag{42}$$

For simplicity, we omit the parameters $\theta_i$ in network $f_i$. Eq.(41) holds due to the linearity of integral and Eq.(42) follows the fact that the sub-network $f_i$ solely depends only on $x_i$ $(i = 1, \ldots, d)$. Without loss of generality, for the $i$-th dimension, we use a piece-wise Gauss–Legendre quadrature rule to compute the one-dimensional integral $\int_{a_i}^{b_i} f_{i,j}(x_i) dx_i$ and then compute $Z_\theta$ in Eq.(42). Specifically, we select $N_i$ Gauss points $\{x_i^{(n_i)}\}_{n_i=1}^{N_i}$ and corresponding weights $\{w_i^{(n_i)}\}_{n_i=1}^{N_i}$, introducing the index $n = (n_1, \cdots, n_d) \in \mathcal{N} := \{1, \ldots, N_1\} \times \cdots \times \{1, \ldots, N_d\}$. Hence, Gauss points and

weights for numerical integration over the high-dimensional domain $\Omega$ are represented as:

$$\{\boldsymbol{x}^{(n)}\}_{n \in \mathcal{N}} = \left\{ (x_1, \cdots, x_d) | x_i = x_i^{(n_i)}, i = 1, \ldots, d \right\}$$
$$\{w^{(n)}\}_{n \in \mathcal{N}} = \left\{ \prod_{i=1}^{d} w_i | w_i = w_i^{(n_i)}, i = 1, \ldots, d \right\} \tag{43}$$

The partition function is computed by numerical integration:

$$
\begin{aligned}
Z_\theta &\approx \int_\Omega \widetilde{p}_\theta(\boldsymbol{x}) d\boldsymbol{x} \\
&\approx \sum_{n \in \mathcal{N}} w^{(n)} \widetilde{p}_\theta(\boldsymbol{x}^{(n)}) \\
&= \sum_{(n_1, \cdots, n_d) \in \mathcal{N}} w_1^{(n_1)} \cdots w_d^{(n_d)} \sum_{j=1}^{r} \prod_{i=1}^{d} f_{i,j}\left(x_i^{(n_i)}\right) \\
&= \sum_{j=1}^{r} \sum_{(n_1, \cdots, n_d) \in \mathcal{N}} \left( w_1^{(n_1)} f_{1,j}\left(x_1^{(n_1)}\right) \right) \cdots \left( w_d^{(n_d)} f_{d,j}\left(x_d^{(n_d)}\right) \right) \\
&= \sum_{j=1}^{r} \left( \sum_{n_1=1}^{N_1} w_1^{(n_1)} f_{1,j}\left(x_1^{(n_1)}\right) \right) \cdots \left( \sum_{n_d=1}^{N_d} w_d^{(n_d)} f_{d,j}\left(x_d^{(n_d)}\right) \right) \tag{44}
\end{aligned}
$$

$\square$

Since the cost of each one-dimensional integral is independent of the dimensionality, the overall computational complexity for the high-dimensional integration is the linear scale of dimension $d$. And we only calculate the partition function once in FPNN.

The advantages and differences between two computations of $Z_\theta$ are summarized as follows:

- **MLP-based FPNN.** The advantages of MLP lie in simpler feature fusion of spatial variables and faster computation of partition functions $Z_\theta$ using MC sampling:

$$Z_\theta = \frac{\nu(\Omega)}{|\mathcal{D}_{\text{norm}}|} \sum_{\boldsymbol{x} \in \mathcal{D}_{\text{norm}}} \widetilde{p}_\theta(\boldsymbol{x}) \tag{45}$$

  However, in particularly high-dimensional settings or the large range of each interval in $\Omega$, the volume of $\Omega$, i.e., $\nu(\Omega) = \prod_{i=1}^{d} |b_i - a_i|$ may exceed machine limits. This issue can be addressed by narrowing the intervals of interest or using higher numerical precision, such as 'float64'.

- **TNN-based FPNN.** TNN achieves high precision in numerical integration for estimating $Z_\theta$, making it more suitable for higher-dimensional problems. Unlike MLP, TNN can maintain appropriate values for the integral in each spatial dimension (i.e., $\int_{a_i}^{b_i} f_{i,j}(x_i) dx_i$), regardless of the dimensionality or the interval length. Introducing regularization terms to guide the learning of smaller unnormalized density is an effective strategy to alleviate the numerical difficulties associated with the partition function.

## C EXPERIMENTAL DETAILS OF PDEs

We remark that FPNN is adaptable to a wide range of SFP equations, but most problems are difficult to obtain analytical solutions for performance evaluation. Inspired by Wang et al. (2024), we use gradient systems to construct test PDE cases with exact solutions. Given any potential function $H(\boldsymbol{x})$ and diffusion term $\boldsymbol{\sigma}(\boldsymbol{x})$, we calculate the diffusion matrix $\boldsymbol{D}(\boldsymbol{x}) = \frac{1}{2}\boldsymbol{\sigma}(\boldsymbol{x})\boldsymbol{\sigma}(\boldsymbol{x})^T$ and set the drift term as follows:

$$\boldsymbol{\mu}(\boldsymbol{x}) = -\boldsymbol{D}(\boldsymbol{x})\nabla H(\boldsymbol{x}) + \nabla \cdot \boldsymbol{D}(\boldsymbol{x}) \tag{46}$$

Table 6: Network structure settings used for visualizing experimental results.

| SFP equations | Model | Layers | Parameters | Steps |
|---|---|---|---|---|
| 4D Ring | TFFN | $4\times[1, 64, 64, 64]$ | 33,792 | $20k$ |
| | TNN | $4\times[8, 64, 128]$ | 35,584 | $10k$ |
| | MLP | $[4, 20, 20, 20, 20]$ | 1,360 | $10k$ |
| 6D Unimodal | TFFN | $6\times[1, 64, 64, 64]$ | 50,688 | $20k$ |
| | TNN | $6\times[5, 64, 64, 64]$ | 52,224 | $5k$ |
| | MLP | $[6, 20, 20, 20]$ | 980 | $20k$ |
| 6D Multi-modal | TFFN | $6\times[1, 64, 64, 64]$ | 50,688 | $20k$ |
| | TNN | $6\times[5, 64, 64, 64]$ | 52,224 | $5k$ |
| | MLP | $[6, 32, 32, 32]$ | 2,336 | $5k$ |
| 10D Multi-modal | TNN | $10\times[1, 64, 64, 64]$ | 84,480 | $1k$ |
| | MLP | $[10, 64, 64, 64]$ | 13,184 | $5k$ |
| Gaussian mixture | TNN | $10\times[1, 64, 64, 64]$ | 126,080 | $5k$ |
| | MLP | $[10, 20, 20, 20, 20]$ | 1,480 | $5k$ |
| Gaussian | TNN | $20\times[1, 64, 64, 64]$ | 252,160 | $3k$ |

Then the solution to the SFP equation $\mathcal{L}p(\boldsymbol{x}) = 0$ with respect to $(\boldsymbol{\mu}, \boldsymbol{D})$ reads

$$p(\boldsymbol{x}) = \frac{1}{Z}e^{-H(\boldsymbol{x})}, \qquad Z = \int e^{-H(\boldsymbol{x})}d\boldsymbol{x} \tag{47}$$

It can be easily verified that

$$\begin{aligned}
\mathcal{L}p(\boldsymbol{x}) &= -\nabla \cdot (p(\boldsymbol{x})\boldsymbol{\mu}(\boldsymbol{x})) + \nabla \cdot [\nabla \cdot (\boldsymbol{D}(\boldsymbol{x})p(\boldsymbol{x}))] \\
&= \nabla \cdot (-p(\boldsymbol{x})\boldsymbol{\mu}(\boldsymbol{x}) + (\nabla \cdot \boldsymbol{D}(\boldsymbol{x}))p(\boldsymbol{x}) + \boldsymbol{D}(\boldsymbol{x})\nabla p(\boldsymbol{x})) \\
&= \nabla \cdot (-p(\boldsymbol{x})\boldsymbol{\mu}(\boldsymbol{x}) + (\nabla \cdot \boldsymbol{D}(\boldsymbol{x}))p(\boldsymbol{x}) - \boldsymbol{D}(\boldsymbol{x})\nabla H(\boldsymbol{x})p(\boldsymbol{x})) \\
&= 0
\end{aligned} \tag{48}$$

The stochastic system is given by $d\mathbf{X} = \boldsymbol{\mu}(\mathbf{X})dt + \boldsymbol{\sigma}(\mathbf{X})d\mathbf{W}_t$. We adopt this form to design SFP equations C.2-C.4. For exact solutions in our experiments, we use the *SymPy* library to compute the partition function in Eq.(47). *SymPy* is a powerful and versatile tool for symbolic mathematics and provides computer algebra system (CAS) capabilities directly in Python. We point out that appropriate simplifications (e.g., completing the square and variable substitution) and the choice of integration order significantly impact the accuracy of results.

In Table 6, we list the network settings used for plotting experimental results, where TNN and MLP represent the architectures in FPNN framework using score PDE loss.

## C.1 4D RING

Consider a stochastic gradient system in four dimensional state space from Zhai et al. (2022). The deterministic part is a gradient system plus a perpendicular rotation term $\boldsymbol{r}(\boldsymbol{x}) = [x_2, -x_1, 0, 0]^T$, where the potential function of the gradient flow is $H(\boldsymbol{x}) = 2(\|\boldsymbol{x}\|_2^2 - 1)^2$. We use the SFP equation (2) with the following drift $\boldsymbol{\mu}(\boldsymbol{x})$ and diffusion $\boldsymbol{D}$:

$$\boldsymbol{\mu}(\boldsymbol{x}) = -\nabla H(\boldsymbol{x}) + \boldsymbol{r}(\boldsymbol{x}), \qquad \boldsymbol{D} = \frac{\sigma^2}{2}\boldsymbol{I}_4, \qquad \boldsymbol{\sigma} = \sigma\boldsymbol{I}_4 \tag{49}$$

Since $\boldsymbol{r}(\boldsymbol{x})$ is orthogonal to the equipotential lines of $H(\boldsymbol{x})$, the rotation term does not change the invariant probability density function. The invariant probability measure has density function

$$p(\boldsymbol{x}) = \frac{1}{Z}e^{-H(\boldsymbol{x})/\sigma^2}, \qquad Z = \pi^2 \int_{-1}^{\infty} (t+1)e^{-2t^2/\sigma^2}dt \tag{50}$$

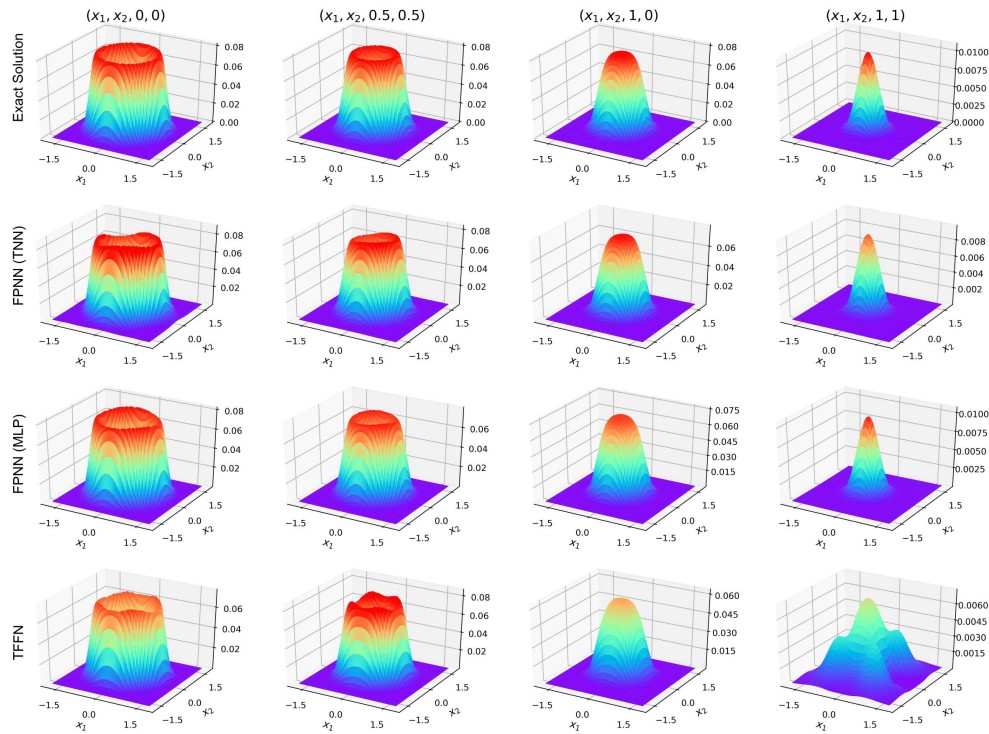

Figure 13: 4D Ring: the exact solution and predicted solutions of different models. First row: the exact solution. Second row: TNN-based FPNN. Third row: MLP-based FPNN. Last row: TFFN.

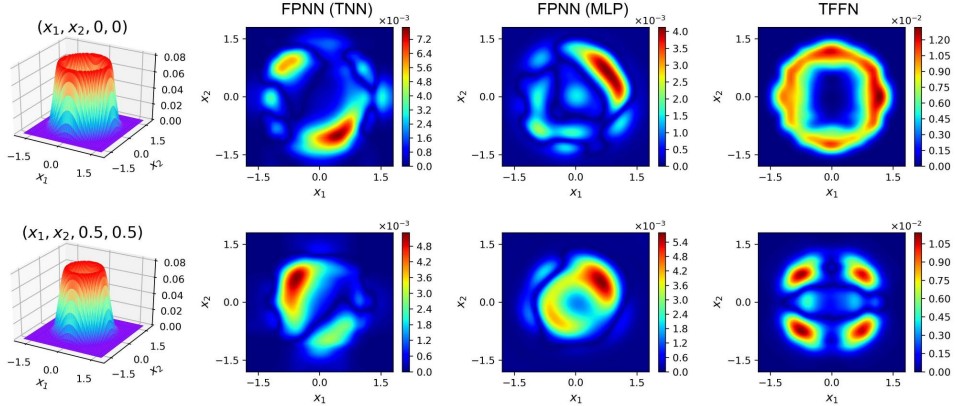

Figure 14: 4D Ring: absolute errors between the exact solution and predicted solutions of different models. First column: the exact solution. Second column: TNN-based FPNN. Third column: MLP-based FPNN. Last column: TFFN.

Here, we utilize the polar coordinates, variable substitution, and the surface area formula of a 4D hypersphere $S = 2\pi^2 r^3$. We set $\sigma = 1$ and calculate the infinite integral for $Z$ using *SymPy* library.

We generated a training dataset $\mathcal{D}_{\text{train}}$ with 20$k$ samples using the 1.5-order SRK method, setting the terminal time $T = 1$, the number of time steps $N = 500$ and a 4-dimensional standard Gaussian distribution $p_0(\boldsymbol{x})$. Based on the range of the dataset $\mathcal{D}_{\text{train}}$, the domain is selected as $\Omega = [-1.8, 1.8]^4$.

For the test dataset generation, we sample 10$k$ initial points uniformly within $\Omega$. Then we perform the gradient ascent using the true solution $p(\boldsymbol{x})$, with a learning rate of $1 \times 10^{-3}$ and the threshold

$\epsilon = 1 \times 10^{-3}$. For fair comparison, we use the same test dataset $D_{\text{test}}$ for TFNN and FPNN with TNN/MLP architectures.

For all SFP equations in our experiments, FPNNs are trained under consistent settings: we use Adam optimizer with a learning rate of 0.01 and a batch size of $2k$, resulting in 10 iterations per epoch. The network structure and prediction performance are detailed in Table 4. TFFN consists of four sub-networks of 3 hidden layers with 64 hidden feature size, updated for $20k$ steps using the Adam optimizer with a learning rate of 0.01, as shown in Table 6. Training data are uniformly sampled from $\Omega$, with $2k$ points resampled per iteration.

## C.2    6D UNIMODAL

In this case, the diffusion $\boldsymbol{D} = \boldsymbol{I}_6$ and the drift $\boldsymbol{\mu}(\boldsymbol{x}) = -\nabla H(\boldsymbol{x})$ take the form in Eq.(46), where the potential function $H(\boldsymbol{x})$ is written as:

$$H(\boldsymbol{x}) = 3((x_1^4 - x_2)^2 + 2x_2^2 + (x_3^4 - x_4)^2 + 2x_4^2 + (x_5^4 - x_6)^2 + 2x_6^2) \tag{51}$$

The variables $(x_1, x_2)$, $(x_3, x_4)$, and $(x_5, x_6)$ have distinct interactions in the form of $(x_i^4 - x_{i+1})^2 + 2x_{i+1}^2, i = 1, 3, 5$. The specific term results in $\boldsymbol{\mu}(\boldsymbol{x})$ being a polynomial function of degree up to 7, introducing significant nonlinearity and challenges for training model. Our proposed PE is inspired by this complexity, and the goal is to alleviate the difficulty of fitting such high-order functions. The density function is more concentrated with steep gradients, and decays rapidly away from the origin. Although the density is unimodal, it remains a challenging problem for solving this SFP equation.

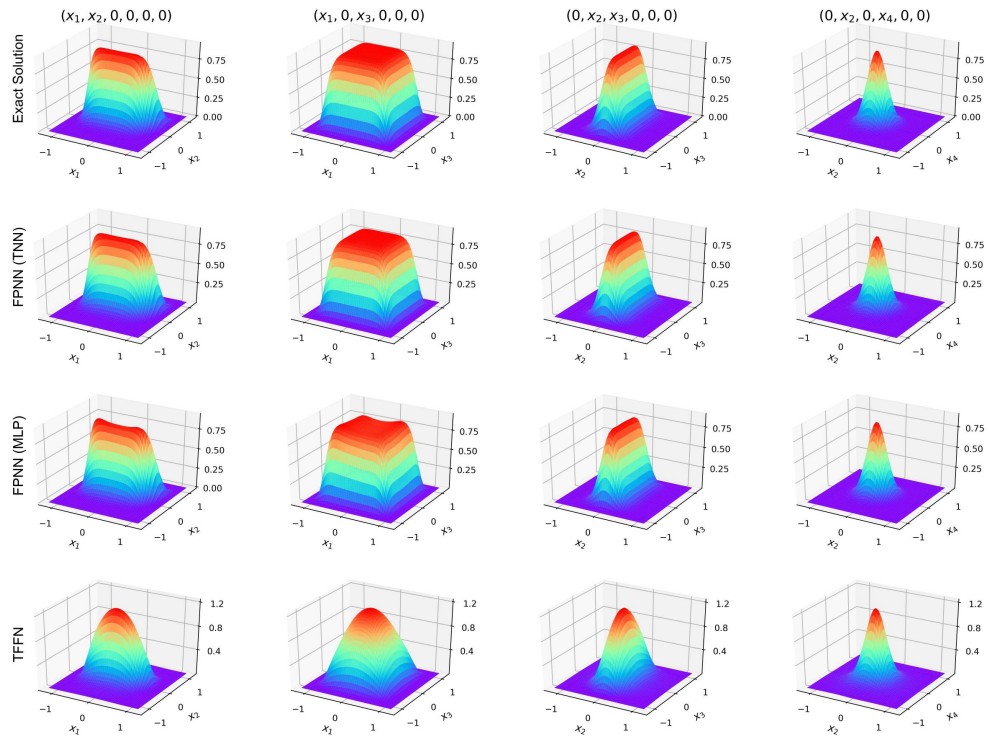

Figure 15: 6D Unimodal: the exact solution and predicted solutions of different models. First row: the exact solution. Second row: TNN-based FPNN. Third row: MLP-based FPNN. Last row: TFFN.

We set the terminal time $T = 1$ and the number of time steps $N = 500$. The initial distribution is a 6-dimensional Gaussian distribution with zero mean and covariance matrix $0.01\boldsymbol{I}_6$. And we generate $20k$ training points and select the domain $\Omega = [-1.2, 1.2]^6$. The test dataset is created by applying gradient ascent to $10k$ initial points from uniform distribution within $\Omega$, with a learning rate of $1 \times 10^{-3}$ and a threshold $\epsilon = 1 \times 10^{-5}$. We use the same training settings for FPNN, and comparisons of different network architectures are shown in Table 5. Notably, PE significantly accelerates the

learning of higher-order densities for TNNs. With appropriate embedding size $m$, TNN achieves MAPE below 10% with fewer steps, while MLP demonstrates its advantage by solving the 6D Unimodal problem with only 980 parameters.

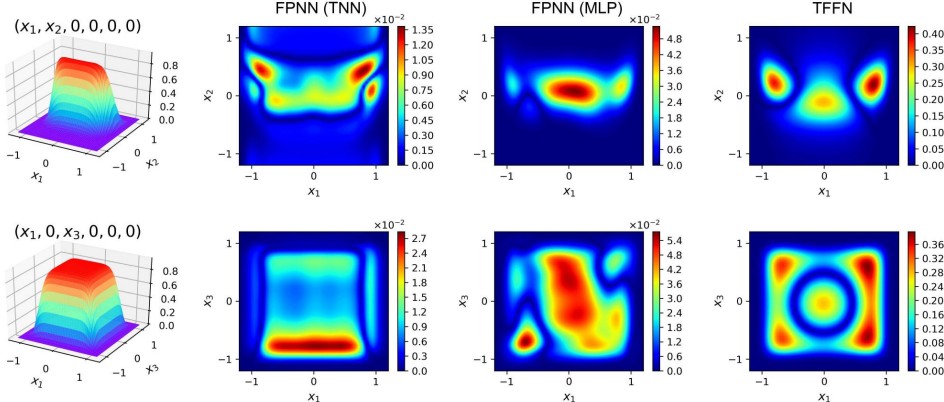

Figure 16: 6D Unimodal: absolute errors between the exact solution and predicted solutions of different models. First column: the exact solution. Second column: TNN-based FPNN. Third column: MLP-based FPNN. Last column: TFFN.

## C.3   6D MULTI-MODAL

We consider the potential $H(\boldsymbol{x})$ with a quadratic form, where the SFP equation is given by the diffusion $\boldsymbol{D} = \boldsymbol{I}_6$ and the drift $\boldsymbol{\mu}(\boldsymbol{x}) = -\nabla H(\boldsymbol{x})$ in Eq.(46). $H(\boldsymbol{x})$ involves interactions among $(x_1, x_2, x_3)$ and $(x_4, x_5, x_6)$, which possesses the following form:

$$
\begin{aligned}
H(\boldsymbol{x}) =& 2(x_1^2 + x_2^2 + x_3^2 + 0.5(x_1 x_2 + x_1 x_3 + x_2 x_3)) - \ln(x_1^2 + 0.02) \\
& - \ln(x_2^2 + 0.02) + 0.5(x_4^2 + x_5^2 + x_6^2 + 0.2(x_4 x_5 + x_4 x_6 + x_5 x_6))
\end{aligned}
\tag{52}
$$

The density function has four modes and we plot the slices of high-density regions from different dimensions to illustrate the performance.

Table 7: 6D Multi-modal: comparison of FPNN with different networks (5$k$ steps).

| Model | Layers | Parameters | MAE | MAPE |
|---|---|---|---|---|
| TNN | [$m$, hidden layers, $r$] | | | |
| | [1, 64, 64, 64] | 50,688 | $4.20 \times 10^{-4}$ | 35.76% |
| | [5, 64, 64, 64] | 52,224 | $3.25 \times 10^{-4}$ | 23.11% |
| | [8, 64, 128] | 53,376 | $2.30 \times 10^{-4}$ | 21.59% |
| MLP | [$d$, hidden layers] | | | |
| | [6, 20, 20, 20] | 980 | $2.00 \times 10^{-4}$ | 12.72% |
| | [6, 32, 32, 32] | 2,336 | $2.05 \times 10^{-4}$ | 12.72% |
| | [6, 64, 64] | 4,608 | $1.98 \times 10^{-4}$ | 12.18% |

We set the terminal time $T = 1$ and the number of time steps $N = 200$. The initial distribution is a 6-dimensional Gaussian distribution with zero mean and covariance matrix $0.1\boldsymbol{I}_6$. We generate $20k$ training points and determine domain $\Omega = [-2, 2]^6$ for 6D Multi-modal problem. We use a a learning rate of $1 \times 10^{-3}$ and a threshold $\epsilon = 1 \times 10^{-5}$ to generate the test dataset with $10k$ samples by gradient ascent method.

From the numerical results in Table 7, MLP outperforms TNN in terms of parameter efficiency, training speed and prediction accuracy, achieving a MAPE of 12%. This may be attributed to the architectural characteristics of TNNs, which is more suitable for the solutions align with the variable separation form and quickly learns discrete low-rank features (see Table 5). Conversely, MLP is

better at fusing features across spatial coordinates in hidden layers. FPNN allows the use of free-form network architectures, providing greater flexibility in handling SFP equations with diverse characteristics.

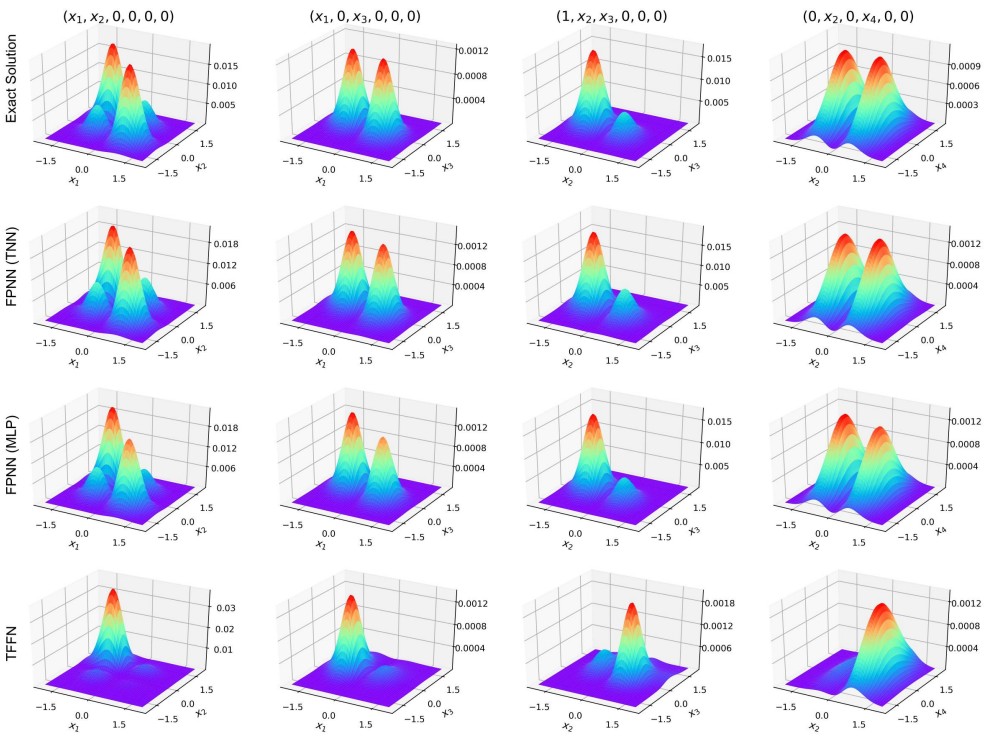

Figure 17: 6D Multi-modal: the exact solution and predicted solutions of different models. First row: the exact solution. Second row: TNN-based FPNN. Third row: MLP-based FPNN. Last row: TFFN.

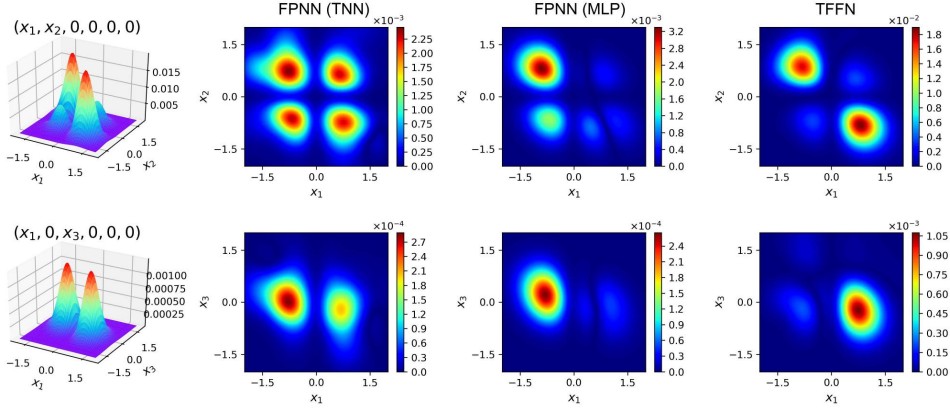

Figure 18: 6D Multi-modal: absolute errors between the exact solution and predicted solutions of different models. First column: the exact solution. Second column: TNN-based FPNN. Third column: MLP-based FPNN. Last column: TFFN.

## C.4   10D MULTI-MODAL

To evaluate the performance in high dimensions, we conduct experiments using a 10-dimensional probability density function with two modes. The SFP equation has the diffusion $D = I_{10}$, and the

potential function $H(\boldsymbol{x})$ for the drift $\boldsymbol{\mu}(\boldsymbol{x})$ as follows:

$$
\begin{aligned}
H(\boldsymbol{x}) =&\, 2.5(x_1^2 + x_2^2 + x_3^2 + 0.1(x_1 x_2 + x_1 x_3 + x_2 x_3)) + 2(x_4^2 + x_5^2 + x_6^2 + 0.2(x_4 x_5 + x_4 x_6 \\
&+ x_5 x_6)) + 3(x_7^2 + x_8^2 - 0.01 x_7 x_8)) + 3(x_9^2 + x_{10}^2 - 0.01 x_9 x_{10})) - \ln(2x_9^2 + 0.02)
\end{aligned}
\tag{53}
$$

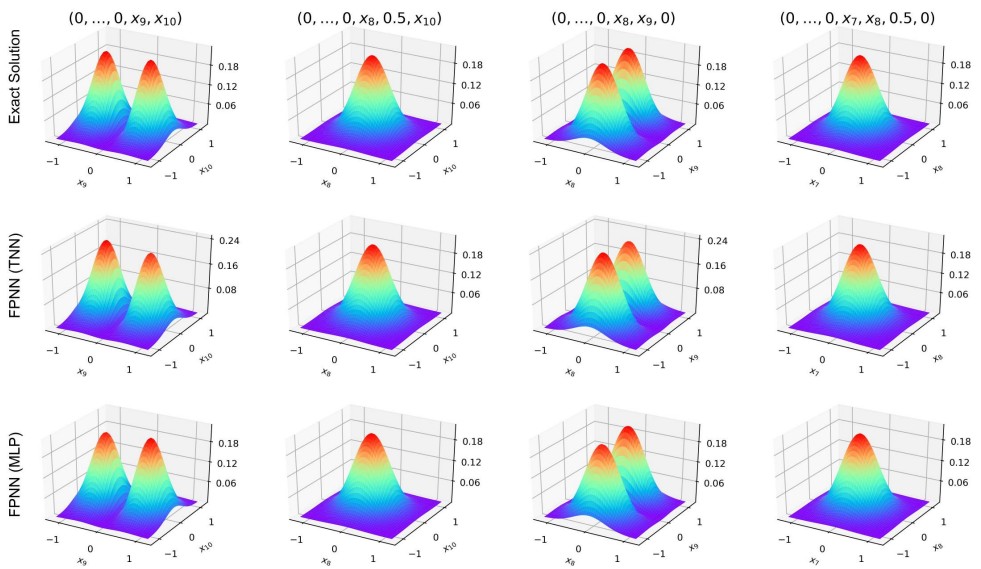

Figure 19: 10D Multi-modal: the exact solution and predicted solutions of FPNN. *Top*: the exact solution. *Middle*: TNN-based FPNN. *Bottom*: MLP-based FPNN.

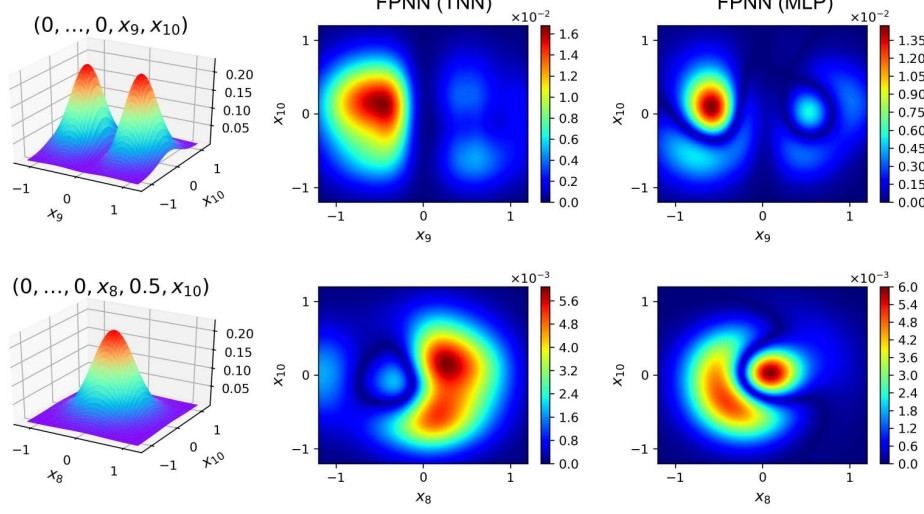

Figure 20: 10D Multi-modal: absolute errors between the exact solution and predicted solutions of different models. *Left*: the exact solution. *Middle*: TNN-based FPNN. *Right*: MLP-based FPNN.

Here, $H(\boldsymbol{x})$ exhibits complex interactions across the coordinates $(x_1, x_2, x_3)$, $(x_4, x_5, x_6)$, $(x_7, x_8)$ and $(x_9, x_{10})$. Due to the component $\ln(2x_9^2 + 0.02)$, there are two peaks along variable $x_9$. By plotting the cross-sectional view of the last two dimensions, we can observe this shape.

We generate a training dataset of $20k$ points using the SRK method, with a terminal time $T = 1$ and $N = 100$ time steps. The initial distribution $p_0(\boldsymbol{x})$ is a 10-dimensional standard Gaussian

distribution and the domain is defined as $[-1.2, 1.2]^{10}$. The test dataset is created using a learning rate of $1 \times 10^{-3}$ and a threshold of $\epsilon = 1 \times 10^{-8}$ to ensure that densities of all $10k$ points exceed $\epsilon$. In FPNN, both TNN and MLP architecture successfully learn the two peaks in the $x_9$ axis, while PINN and TFFN fail to capture this feature. It is evidenced by the divergence between model predictions and true solutions for TFFN in Figure 4. The normalization condition influence the representational capacity and optimization process of models, as discussed in Sec. 1. This results in poor performance of existing methods for 4-6 dimensional SFP equations and failure in higher-dimensional problems.

## C.5    10D GAUSSIAN MIXTURE

Following the test problem with two peaks (Tang et al., 2022), we construct a Gaussian mixture distribution $p(\boldsymbol{x}) = \beta_1 p_1(\boldsymbol{x}) + \beta_2 p_2(\boldsymbol{x})$, comprised of two 10-dimensional Gaussian distributions $p_1$ and $p_2$. For $k = 1, 2$, each $p_k(\boldsymbol{x})$ is defined by a probability density function with mean $\mu_k$ and covariance matrix $\Sigma_k$:

$$\beta_1 = 0.55, \quad \mu_1 = [-1.5, -0.8, 1.3, 0.2, -0.1, 0, 0, 0, 0, 0], \quad \Sigma_1 = \begin{pmatrix} \Sigma_{11} & & & \\ & \Sigma_{12} & & \\ & & \Sigma_{13} & \\ & & & \Sigma_{14} \end{pmatrix}$$

$$\Sigma_{11} = \begin{pmatrix} 2.2 & 0 & 0 \\ 0 & 1.2 & 0 \\ 0 & 0 & 2 \end{pmatrix}, \ \Sigma_{12} = \begin{pmatrix} 1.5 & 0.2 & 0.4 \\ -0.1 & 1.2 & 0.4 \\ -0.2 & -0.2 & 0.8 \end{pmatrix}, \ \Sigma_{13} = \begin{pmatrix} 0.4 & 0.3 \\ 0.3 & 0.9 \end{pmatrix}, \ \Sigma_{14} = \begin{pmatrix} 1 & 0 \\ 0 & 1 \end{pmatrix} \tag{54}$$

$$\beta_2 = 0.45, \quad \mu_2 = [1.2, 1, -1.5, 0, 0, 0.1, 0, 0, 0, 0], \quad \Sigma_2 = \begin{pmatrix} \Sigma_{21} & & & \\ & \Sigma_{22} & & \\ & & \Sigma_{23} & \\ & & & \Sigma_{24} \end{pmatrix}$$

$$\Sigma_{21} = \begin{pmatrix} 2.2 & 0 & 0 \\ 0 & 1 & 0 \\ 0 & 0 & 1.5 \end{pmatrix}, \ \Sigma_{22} = \begin{pmatrix} 1.2 & 0.4 & -0.2 \\ -0.4 & 1.2 & -0.3 \\ 0.2 & -0.1 & 1.2 \end{pmatrix}, \ \Sigma_{23} = \begin{pmatrix} 0.8 & 0 \\ 0 & 0.3 \end{pmatrix}, \ \Sigma_{24} = \begin{pmatrix} 1 & 0 \\ 0 & 1 \end{pmatrix} \tag{55}$$

The matrices $\Sigma_1$ and $\Sigma_2$ are positive definite. FPNN remains applicable for more randomly constructed Gaussian components. To improve visualization, we make some modifications to the mean $\mu_1$, $\mu_2$. First, two Gaussian peaks are placed at a certain distance apart and have different relative positions on the $(x_1, x_2)$ and $(x_1, x_3)$ planes. Secondly, we set the last seven components of the mean near zero, so that we examine the cross-sections $(x_1, x_2, 0, \ldots, 0)$ and $(x_1, 0, x_3, 0, \ldots, 0)$ to reflect the high-density regions of exact solution. It is straightforward to verify that $p(\boldsymbol{x})$ serves as the true solution to following SFP equation:

$$-\nabla \cdot (p(\boldsymbol{x}) \nabla \log(\beta_1 p_1(\boldsymbol{x}) + \beta_2 p_2(\boldsymbol{x}))) + \nabla^2 p(\boldsymbol{x}) = 0 \tag{56}$$

where we get the corresponding drift $\boldsymbol{\mu}(\boldsymbol{x}) = \nabla \log(\beta_1 p_1(\boldsymbol{x}) + \beta_2 p_2(\boldsymbol{x}))$ and diffusion $\boldsymbol{D} = \boldsymbol{I}_{10}$.

As the region expands, the stochastic system takes longer to reach a steady state. We select a terminal time of $T = 5$ and $N = 500$ time steps. $20k$ data points are sampled from a 10-dimensional standard Gaussian distribution to serve as the initial set, followed by the SRK method to obtain the target distribution $p_T$ and domain $\Omega = [-5, 5]^{10}$. The test dataset with $10k$ points is sampled from two Gaussian distributions according to weights.

Notably, even though the true solution has the order of $10^{-5}$, simple network architectures such as TNN and MLP successfully learn the two Gaussian peaks located on the coordinate planes $(x_1, x_2, 0, \ldots, 0)$ and $(x_1, 0, x_3, 0, \ldots, 0)$ at 600 and 400 steps, ultimately achieving a mean relative error of $1.86 \times 10^{-7}$ and $1.16 \times 10^{-7}$, respectively. And MLP reaches a MAPE of 18.38% with only 1,480 parameters. The final prediction results and absolute errors are presented in Figure 21 and Figure 22.

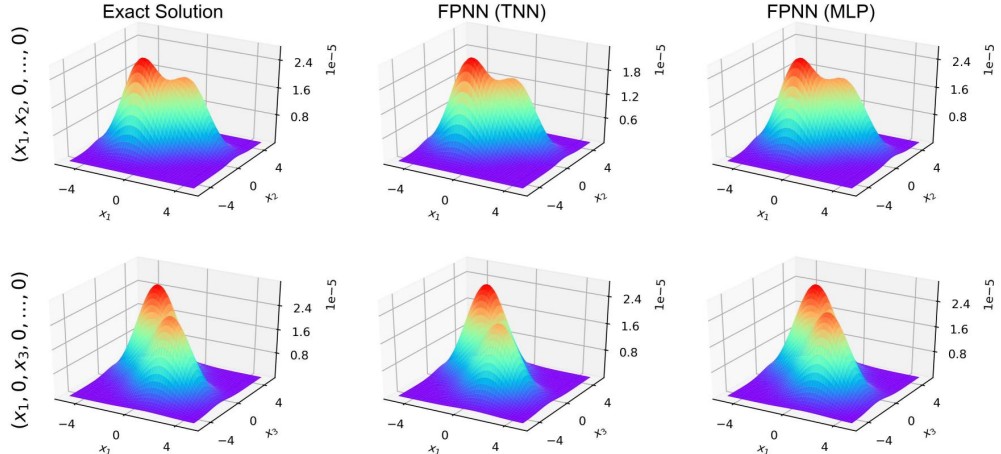

Figure 21: 10D Gaussian mixture: the exact solution and predicted solutions of FPNN. *Left*: the exact solution. *Middle*: TNN-based FPNN. *Right*: MLP-based FPNN.

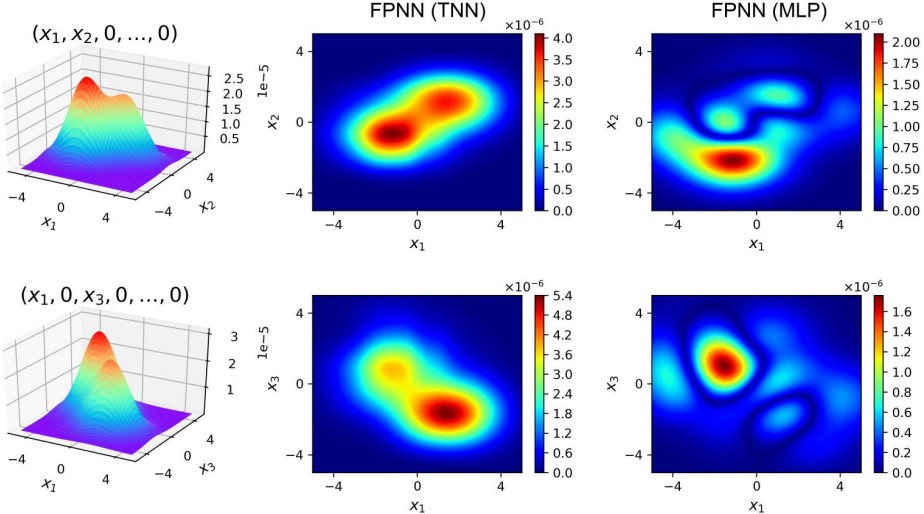

Figure 22: 10D Gaussian mixture: absolute errors between the exact solution and predicted solutions of different models. *Left*: the exact solution. *Middle*: TNN-based FPNN. *Right*: MLP-based FPNN.

## C.6 20D GAUSSIAN

Finally, we consider a SFP equation with drift term $\boldsymbol{\mu}(\boldsymbol{x}) = -a\boldsymbol{x}$ and diffusion matrix $\boldsymbol{D} = \frac{\sigma^2}{2}\boldsymbol{I}_d$. The exact solution is expressed as follows:

$$p(\boldsymbol{x}) = \left(\frac{a}{\pi\sigma^2}\right)^{d/2} \exp\left(-\frac{a\|\boldsymbol{x}\|_2^2}{\sigma^2}\right) \tag{57}$$

In our experiments, we select $d = 20$, $a = 3$ and $\sigma = 1.5$ to test our FPNN with TNN network. So the true PDF is a 20-dimensional Gaussian distribution with zero mean and covariance matrix $\frac{\sigma^2}{2a}\boldsymbol{I}_{20}$. The training dataset is generated by the SRK method, yielding $20k$ samples with $T = 1$ and $N = 100$. Since the initial distribution is chosen as a 20-dimensional standard Gaussian distribution, the drift effect towards the origin dominates, causing particles to converge towards $\Omega = [-2, 2]^{20}$ during the SDE simulation. The test dataset is directly constructed by sampling $10k$ points from the true distribution.

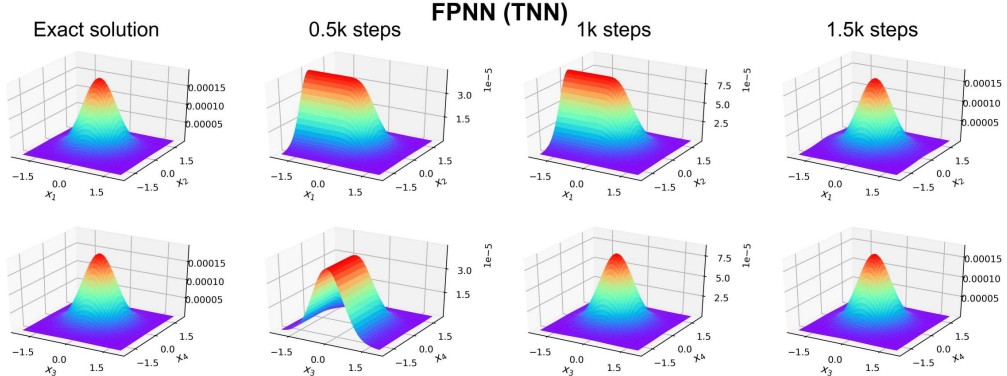

Figure 23: 20D Gaussian: the predicted solution of FPNN during training.

After $1.5k$ update steps, FPNN successfully learns a 20-dimensional Gaussian distribution, achieving a MAPE of 23.29%. As shown in Figure 10, each dimension aligns well with the true solution, demonstrating both the effectiveness of our method and the validity of evaluation metrics.

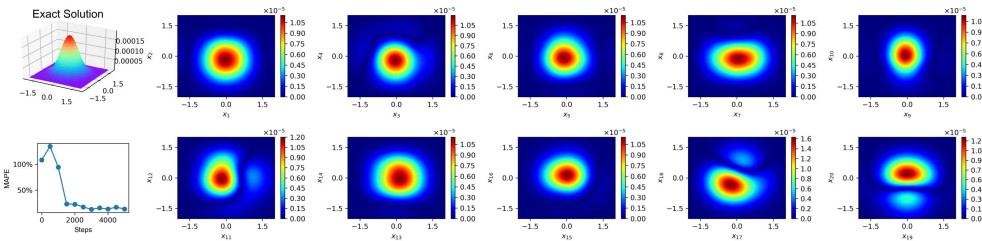

Figure 24: 20D Gaussian: absolute errors between the exact solution and predicted solution of FPNN. The first column shows the exact solution and the changes of MAPE during training. The remaining parts display errors for pairs of adjacent dimensions, such as $(x_1, x_2, 0, \ldots, 0)$, $(0, 0, x_3, x_4, 0, \ldots, 0)$, and so on.

