# OpenReview forum: "Score-based free-form architectures for high-dimensional Fokker-Planck equations"
_ICLR.cc/2025/Conference — ICLR 2025 Poster_

### Official Review · Reviewer_eksf · 2024-10-24

**Soundness:** 3
**Presentation:** 3
**Contribution:** 3
**Rating:** 8
**Confidence:** 4

**Summary:**

This paper intends to solve high-dimensional Fokker-Planck equations which faces several challenges including curse of dimension, normalization constraint, etc. The solution proposed is to train neural network with a score-matching loss which bypasses normalization constraint by computing normalization constant as post-process. The method belongs to supervised learning, where training data is generated by stochastic Runge-Kutta method. The result seems effective and surpasses baseline model in accuracy.

**Strengths:**

**Originality** The score-based loss is novel and seemly interesting. Given the close connection between Fokker-Planck (FP) equations and diffusion process and the noticeable succuss of diffusion model with score-matching loss, it is worthy trying to solve FP with score-based loss.

**Clarity** I find the paper very clear to read and well organized.

**Weaknesses:**

At the first glance, it is a seemly natural and attractive idea to solve Fokker-Planck (FP) equations with the proposed score-matching loss, especially considering the success in training diffusion models and the close connection between stochastic process and FP equations. However, after more careful thoughts I find it hard to reason through the following questions:

1. If the proposed method needs a postprocess of calculating normalizing constant, why not treating PINN with the same postprocess? One of the major motivations of the paper is to deal with normalization condition (NC), which the authors criticized PINN being hard to satisfy with soft constraints. However, if PINN is obtained without NC and is normalized with the same quadrature technique afterwards, this motivation is weakened.

2. The score-based FP loss (equation (6)) is derived for static FP equations. What is the difficulty with non-stationary FP? It seems to me the residual loss can be transformed similarly and just one more term is needed in equation (15), which is $\partial_t \log p_{\theta}(x)$. Even if we only consider SFP, it is clear now that score-based FP loss is essentially equivalent to residual loss of PINN. Therefore, I wonder what benefit score-based FP loss can introduce?

Based on these questions, I suggest the authors do an ablation study of replacing score-based FP loss with PINN loss (residual loss) without normalization constraint. Otherwise, the improvement of FPNN over TFFN may be purely due to removing normalization constraint from loss function.

**Questions:**

See weaknesses above. Also, when using MAPE for metric, how do you calculate $p(x)$ for test dataset?

---

> ### Author Response · Authors · 2024-11-14
>
> Thank you very much for your comments and constructive suggestions. There are some key points that may require further clarification to enhance your understanding of the improvements in our work.
>
> $\textbf{Weakness 1}$
>
> PINNs inherently do not support a postprocess of calculating the normalizing constant, as the objective function is defined as follows: (plain PDE loss)
> $$\mathcal{J}\_{\text{plain}}(p\_\theta)=\mathbb{E}[|\nabla\cdot(p\_\theta\mu)-\nabla\cdot(\nabla\cdot(Dp_\theta))|]$$
> It is evident that the zero solution ($p_\theta = 0$) also satisfies this equation. Therefore, directly minimizing the loss function above often leads to network collapse, where $p_\theta$ converges to a trivial solution. Existing strategies generally fall into two categories involving normalization constraints:
> 1. Adding a penalty term $\mathcal{J}\_{\text{norm}}=\left(\frac{|\Omega|}{|\mathcal{D}|}\sum\_{x\in\mathcal{D}} p\_\theta(x) - 1\right)^2$, where the dataset $\mathcal{D}$ is sampled uniformly from $\Omega$.
> 2. Using specific structures to represent the density function.
>
> However, the first method requires manual balancing of multi-objective loss function, while the second method may limit the model's representation capacity (Fig.1).
>
> In contrast, our method deviates from all existing approaches by changing the loss function directly. We introduce a new loss function as: (score PDE loss)
> $$\mathcal{J}\_{\text{score}}(s\_\theta)=\mathbb{E}\left[|\nabla\log \widetilde{p\_\theta}(x) \cdot \widetilde{\mu}(x) + \nabla\cdot\widetilde{\mu}(x)|\right], \quad \widetilde{\mu}(x) = \mu(x) - \nabla\cdot D(x) - D(x)\nabla\log \widetilde{p\_\theta}(x)$$
> In this form, the trivial solution $p_\theta = 0$ no longer satisfies the equation. The *score PDE loss* retains equivalence with the *plain PDE loss* while avoiding a zero solution and eliminating the need for handling normalization conditions in training process.
>
> To further illustrate, consider a special case where $D(x) = I_d$. If minimizing the score loss $\mathcal{J}_{\text{score}}$ drives $\widetilde{\mu}(x)$ to zero, the training process becomes equivalent to optimizing the objective function of *flow matching*:
> $\mathbb{E}\left[\|\mu(x) - \nabla\log \widetilde{p\_\theta}(x)\|^2\right]$. Thus, our method inherently integrates score matching while adhering to the original Fokker-Planck equation and corresponding physical laws.
>
> Therefore, it is natural to perform a postprocess of calculating the partition function $Z_\theta$ under the score PDE loss. The advantages are also clear: it reduces computational costs, eliminates the interference caused by normalization constraints on optimization dynamics, and ensures a more efficient and coherent training process.
>
> $\textbf{Weakness 2}$
>
> Regarding the non-stationary Fokker-Planck (time-dependent FP, TFP) equation, your understanding is indeed correct. The transformation remains applicable to TFP equations, with one more term $\partial_t \log p_\theta(x, t)$. However, our target is to completely decouple the normalization condition from training process and instead use a neural network to model $\widetilde{p_\theta}$. It is important to note that $\log p_\theta(x, t) = \log \widetilde{p_\theta}(x, t) - \log Z_\theta(t)$, which implies that the loss function would introduce a term $\partial_t \log \widetilde{p_\theta}(x, t) - \partial_t \log Z_\theta(t)$, necessitating the evaluation of the partition function. And the loss function is not merely in terms of $\nabla \log p(x, t)$ (i.e., the score form).
>
> Moreover, there is an additional technical challenge: for non-localized density functions, identifying a suitable integration domain $\Omega$ at any given time $t$ to compute the time-varying normalizing function $Z_\theta(t)$ is challenging. It may be feasible to calculating the discrete sequence $\\{Z_1, Z_2, \ldots, Z_N\\}$ at time steps $\\{t_1, t_2, \ldots, t_N\\}$ and then interpolating to obtain a continuous function $Z_\theta(t)$.
>
> In the future work, we aim to further explore and refine the score-based algorithm for TFP equations.

---

> > ### Author Response · Authors · 2024-11-14
> >
> > $\textbf{Suggestion in Weakness}$
> >
> > With the reformulation and clarification of $\mathcal{J}\_{\text{plain}}$ and $\mathcal{J}\_{\text{score}}$, we can find that the comparison between the TNN-based FPNN and TFFN in our experiments essentially aligns with the ablation study you suggested. We controlled for the network architecture, spatial domain, number of training points, optimizer, learning rate, and other hyperparameters to remain consistent, with the only difference being the replacement of the PINN loss (residual loss) with our score-based FP loss. Using the same test dataset $\mathcal{D}_\text{test}$ (which is uniformly sampled from $\Omega$ and not used during training for either model), we observed that FPNN significantly outperformed TFFN in terms of training speed, computational costs and evaluation metrics (MAE and MAPE).
> >
> > $\textbf{Questions}$
> >
> > In our experiments, SFP problems are constructed with potential functions for which analytical solutions are available to evaluate model performance (see Eq.(23) in Appendix C). Additionally, we did not randomly sample the test points from the entire space, as it could result in very small true values for density (often happens in high-dimensional settings) and render MAPE ineffective. Instead, we use the gradient ascent method on the analytical solution $p(x)$ to ensure that all densities of test data exceeds a certain threshold $\epsilon$. We then record these spatial points $x$ along with their corresponding densities $p$ as test dataset.
> >
> > If you have any further questions or suggestions, please feel free to give comments or remarks. We are open to a more in-depth discussion on both static FP (SFP) and non-stationary FP (TFP) equations.

---

> ### Comment · Reviewer_eksf · 2024-11-15
>
> Thank you for answering my questions. Now I have a better grasp of the overall idea of this paper. I was impressed by the idea of connecting score matching with solving FP equations, and now I'm convinced it is also more advantageous over PINN. I will change my score when permission is given to me and recommend for **accept**. To update my evaluation:
>
> Soundness: 3: good
>
> Presentation: 3: good
>
> Contribution: 3: good
>
> The main contribution of this paper is to solve high-dimensional Fokker-Planck equations by score matching loss, which has two advantage that PINNs do not possess:
> 1. Leveraging the data sampling method of diffusion process to generate data, which is latter used in score matching loss. And I do not see how such data can be easily integrated into PINN loss, but it is natural to use the data for score-matching loss.
> 2. Circumventing the normalization condition in loss function which causes difficulty in training. Instead, the paper proposed calculating normalizing constant by postprocessing.

---

> > ### Author Response · Authors · 2024-11-15
> >
> > Thank you for your insightful and kind comments! We are glad to hear that you recognize and appreciate our ideas.
> >
> > We fully agree with your second point. You have accurately and succinctly summarized the key contributions of our work. As for the first point, there are several studies on adaptive sampling methods for PINNs [1-6], which usually involve resampling or adding training points based on the PDE residuals. Thus, data from the diffusion process could also be used to compute the PINN loss $\mathcal{J}_{\text{plain}}$. But it does not fundamentally address the core challenges in solving high-dimensional Fokker-Planck equations and is less natural than score matching.
> >
> > We also recommend that you consider the numerical benefits of FPNN over PINN in learning high-dimensional densities (*Advantage 2* in our introduction). For instance, in the *10D Gaussian Mixture* problem, the magnitude of the true solution is around $10^{-5}$. If PINN is trained using the objective function $\mathcal{J}\_{\text{plain}} + \lambda \mathcal{J}\_{\text{norm}}$ with an MLP architecture, it is challenging for the model to produce outputs of that magnitude, especially without labeled data or prior knowledge for post-standardization.
> >
> > In contrast, FPNN leverages a score-based model, allowing the network to freely choose an appropriate scale and learn the unnormalized density $\widetilde{p_\theta}$. By calculating normalizing constant $Z_\theta$ through a post-processing step, the model finally yields $p_\theta(x) = \widetilde{p_\theta}(x) / Z_\theta$, thereby enabling outputs on the scale of $10^{-5}$. This significantly mitigates the numerical difficulties inherent in high-dimensional Fokker-Planck equations.
> >
> > **References**
> >
> > [1] @article{lu2021deepxde,
> >   title={DeepXDE: A deep learning library for solving differential equations},
> >   author={Lu, Lu and Meng, Xuhui and Mao, Zhiping and Karniadakis, George Em},
> >   journal={SIAM review},
> >   volume={63},
> >   number={1},
> >   pages={208--228},
> >   year={2021},
> >   publisher={SIAM}
> > }
> >
> > [2] @article{nabian2021efficient,
> >   title={Efficient training of physics-informed neural networks via importance sampling},
> >   author={Nabian, Mohammad Amin and Gladstone, Rini Jasmine and Meidani, Hadi},
> >   journal={Computer-Aided Civil and Infrastructure Engineering},
> >   volume={36},
> >   number={8},
> >   pages={962--977},
> >   year={2021},
> >   publisher={Wiley Online Library}
> > }
> >
> > [3] @article{wang20222,
> >   title={Is $L^2$ Physics Informed Loss Always Suitable for Training Physics Informed Neural Network?},
> >   author={Wang, Chuwei and Li, Shanda and He, Di and Wang, Liwei},
> >   journal={Advances in Neural Information Processing Systems},
> >   volume={35},
> >   pages={8278--8290},
> >   year={2022}
> > }
> >
> > [4] @article{wu2023comprehensive,
> >   title={A comprehensive study of non-adaptive and residual-based adaptive sampling for physics-informed neural networks},
> >   author={Wu, Chenxi and Zhu, Min and Tan, Qinyang and Kartha, Yadhu and Lu, Lu},
> >   journal={Computer Methods in Applied Mechanics and Engineering},
> >   volume={403},
> >   pages={115671},
> >   year={2023},
> >   publisher={Elsevier}
> > }
> >
> > [5] @article{hou2023enhancing,
> >   title={Enhancing PINNs for solving PDEs via adaptive collocation point movement and adaptive loss weighting},
> >   author={Hou, Jie and Li, Ying and Ying, Shihui},
> >   journal={Nonlinear Dynamics},
> >   volume={111},
> >   number={16},
> >   pages={15233--15261},
> >   year={2023},
> >   publisher={Springer}
> > }
> >
> > [6] @article{tang2023adversarial,
> >     title={Adversarial Adaptive Sampling: Unify {PINN} and Optimal Transport for the Approximation of {PDE}s},
> >     author={Kejun Tang and Jiayu Zhai and Xiaoliang Wan and Chao Yang},
> >     journal ={The Twelfth International Conference on Learning Representations},
> >     year={2024}
> > }

---

> > > ### Comment · Reviewer_eksf · 2024-11-18
> > >
> > > Thanks for clarifying the first point. Now I see the connection between adaptive sampling and score-matching loss. I have updated the official score as well.

---

> > > > ### Author Response · Authors · 2024-11-18
> > > >
> > > > Thanks for your nice comments! It is of great significance to our work.

---

### Official Review · Reviewer_hrfk · 2024-10-28

**Soundness:** 3
**Presentation:** 3
**Contribution:** 3
**Rating:** 6
**Confidence:** 2

**Summary:**

The paper introduces the Fokker-Planck Neural Network (FPNN) framework, a novel approach to solving high-dimensional steady-state Fokker-Planck equations by leveraging a score-based PDE loss that decouples density normalization from score learning. This decoupling allows FPNN to avoid continuous computation of the partition function, enhancing computational efficiency and stability, particularly for complex high-dimensional problems. Experimental results demonstrate FPNN’s superior performance over existing methods, achieving high accuracy and significant speedups, making it a promising candidate for scalable and efficient solutions to Fokker-Planck equations.

**Strengths:**

(1) This paper proposes a novel network for solving high-dimensional steady-state Fokker-Planck equations. By utilizing a score-based PDE loss that decouples score learning from density normalization, the network achieves an effective balance between representational capacity and the constraints required for accurate solutions.

(2) The performance results are promising, though further validation is needed to strengthen the findings.

(3) The presentation of this paper is good.

**Weaknesses:**

(1) In this paper, using SRK as part of the data generation process for steady-state Fokker-Planck equations is effective, but it might lead to an "unfair advantage" in comparisons if other baseline methods do not leverage a similar approach for handling randomness or approximating steady states. Thus, it will be meaningful to test the performance of the proposed method with the same training data as proposed in TFFN paper.

(2) Score-based methods have shown strong performance for in-distribution problems, but they often suffer from significantly reduced effectiveness on out-of-distribution (OOD) tasks. I am curious whether the authors evaluated the OOD performance for this model. Additionally, as mentioned previously, I wonder if the data generation approach used in this study simplified the distribution, making it easier for the network to learn. If that is the case, the improvements might be attributed more to the engineering aspects of data preparation rather than advancements in the network architecture itself.

(3) Additional comparisons would strengthen this paper. Although FPNN outperforms TFFN in the results presented, I could not find any published reference for TFFN, suggesting it may only be available on arXiv and has not yet undergone peer review. To more convincingly demonstrate the effectiveness of the proposed method, I recommend that the authors include additional, widely recognized baseline methods. This would provide a more comprehensive evaluation of FPNN’s efficiency and robustness.

**Questions:**

The questions are addressed in the weaknesses section. Although this research is not fully aligned with my area of expertise, I will follow the rebuttal process and hope that my suggestions help improve the clarity and presentation of the paper.

---

> ### Author Response · Authors · 2024-11-19
>
> Thanks for your thoughtful comments and constructive suggestions. We address the fundamental differences between FPNN and existing methods, as well as the motivation behind our score PDE loss in the **Background** and **Our Work** sections of Our **Official Comments**. We believe the score loss is a key factor in the superior performance of our model and these sections provide an interesting story. Considering that our work may not fully align with your research interests or expertise, we would like to clarify our PDE task further.
>
> **Our Task**
>
> Our goal is to solve an important class of PDE systems: the steady-state Fokker-Planck (SFP) equation. Unlike other PDEs, its solution represents the density function of an invariant equilibrium distribution, and needs to satisfy additional normalization constraints.
>
> Due to the high dimensionality, we want to leverage deep learning methods to find the solution to a given SFP equation. It means that the drift $\mu(x)$ and diffusion $D(x)$ are known in advance and we want to know which steady state the system reaches under both drift and diffusion effects.
>
> $\textbf{Weakness 1}$
>
> If you understand the issues inherent in the original loss function, $\mathcal{J}_{\text{plain}}$, it becomes evident that the advantages of FPNN stem entirely from the newly introduced **score PDE loss**, rather than from the training data or network architecture.
>
> - **Data.** Data from SDE simulations help define the scope of this stochastic system, because not all SFP equations have density concentrated near the origin. And before solving the PDEs, we have no idea about the distribution of solutions. One major effect of generating data using the SRK method is to provide a reasonable estimation of  domain.
> - **Network Architecture.** The flexibility in network design is another benefit brought about by the score PDE loss, which totally decouples the normalization condition. This allows us to employ unrestricted network architectures while still strictly satisfying the normalization condition.
>
> In our current experiments, we use datasets $\mathcal{D}\_{\text{train}}$ (generated via SRK method) and $\mathcal{D}\_{\text{uniform}}$ (uniformly sampled on $\Omega$) for $\mathcal{J}\_{\text{score}}$ (FPNN) and $\mathcal{J}\_{\text{plain}}$ (TFFN), respectively, to ensure consistency in the statistical expectations within the loss functions. We acknowledge that this might have caused some confusion.
>
> Nevertheless, we are confident that regardless of whether both use $\mathcal{D}\_{\text{train}}$ or $\mathcal{D}\_{\text{uniform}}$, the training results would still demonstrate the superiority of $\mathcal{J}\_{\text{score}}$ over $\mathcal{J}\_{\text{plain}}$. This is due to the fundamental differences in the optimization dynamics of the two objective functions. We will include additional experimental results to validate this claim and update the manuscript and supplementary codes accordingly in few days. This should address concerns about the possibility of an "unfair advantage" for FPNN over TFFN.
>
> **Weakness 2**
>
> As outlined in the **Our Task** section, our primary focus is on the steady-state distribution of a stochastic system under the interactions between drift and diffusion. Thus, we focus solely on the solution to the current system, (i.e., the invariant equilibrium distribution) and do not address OOD tasks. Our main concern is the applicability of FPNN, specifically whether it can efficiently solve more types of SFP equations and accurately compute the density function under various complex drift and diffusion dynamics.
>
> **Weakness 3**
>
> The comparison with existing state-of-the-art methods, as discussed in the **Baselines** section of the **Official Comments**, highlights the challenges posed by high-dimensional problems. These challenges render most existing methods ineffective, preventing them from correctly solving our PDE cases. We adopt a completely different approach, improving the loss function and optimization dynamics, which enables us to accurately learn the density function across a wide range of high-dimensional problems.
>
> If you have any further questions or suggestions, please feel free to give comments or remarks. We are open and appreciate a more in-depth discussion.

---

### Official Review · Reviewer_aoGx · 2024-11-03

**Soundness:** 3
**Presentation:** 3
**Contribution:** 2
**Rating:** 6
**Confidence:** 3

**Summary:**

This paper introduces the Fokker-Planck Neural Network (FPNN), a novel framework for solving high-dimensional steady-state Fokker-Planck (SFP) equations. Traditional deep learning approaches to these equations face challenges with representation capacity, loss function balancing, and maintaining normalization constraints. The proposed FPNN addresses these issues by decoupling score learning from density normalization, using a score PDE loss that enables strict adherence to normalization constraints while allowing flexible, mesh-free architectures. FPNN achieves significant computational efficiency, with over a 20x speedup compared to state-of-the-art methods, and requires only minimal parameters for high accuracy. The authors demonstrate its effectiveness on 4D, 6D, and even 20D problems, achieving low relative errors without labeled data. The FPNN framework contributes a fast, efficient, and accurate solution method for high-dimensional Fokker-Planck equations, with applications in computational physics and related fields.

**Strengths:**

The paper presents a novel Fokker-Planck Neural Network (FPNN) that innovatively decouples score learning and density normalization for high-dimensional steady-state Fokker-Planck (SFP) equations. Traditional deep learning methods for Fokker-Planck equations generally incorporate the PDE residual as part of the loss function but often encounter issues with representation capacity, balancing multi-objective loss functions, and satisfying normalization constraints. FPNN addresses these challenges by using a score PDE loss, which separates score learning from normalization, allowing for a flexible, mesh-free network architecture. This approach is original as it removes the dependency on specific architectures, enables strict normalization through a single computation of the partition function, and offers substantial computational gains over existing methods. By rethinking how neural networks approach Fokker-Planck equations, the authors introduce a framework that improves upon limitations of previous methods, potentially broadening the scope of high-dimensional PDE applications in machine learning.

**Weaknesses:**

1. Limited Discussion on Practical Constraints and Limitations
2. Lack of Scalability and Computational Complexity Analysis
3. Lack of Comparison with Other Recent Advances

**Questions:**

1. Could be the proposed method widely used for other PDEs?
2. Please compare with recent developed methods.

---

> ### Author Response · Authors · 2024-11-19
>
> Thank you very much for comments.
>
> $\textbf{Question 1}$
>
> The goal of our work is to solve an important class of PDE systems: the steady-state Fokker-Planck (SFP) equation. Its solution represents the density function of an invariant equilibrium distribution and must satisfy additional normalization constraints (NC). Unlike other PDEs, NC introduces significant optimization difficulties and results in the quite small solution in high-dimensional spaces and numerical challenges. Therefore, our work focuses on using a score loss to better handle the normalization condition and addressing this specific type of SFP equation with clear physical significance.
>
> $\textbf{Question 2 and Weakness 3}$
>
> We provide a comparison with state-of-the-art methods in the **Baselines** section of the **Official Comments**. Additionally, in the **Background** section, we compare and classify lots of deep-learning approaches with FPNN. Basically all existing methods can be classified into these ideas, demonstrating the novelty and effectiveness of our method from a theoretical perspective.
>
> $\textbf{One more thing for Question 1}$
>
> Although FPNN does not address other PDEs, as the rebuttal process has progressed and our understanding of $\mathcal{J}\_{\text{plain}}$ and $\mathcal{J}\_{\text{score}}$ has deepened, we believe that the phenomenon of minimizing the PDE loss leading to trivial solutions may globally or locally exist in many PDEs. For instance:
>
> **Burgers equation：**$u_t+uu_x=vu_{xx}$
>
> **KdV equation：**$u_t+6uu_x+u_{xxx}=0$
>
> **Schrödinger equation：**$ih_t+0.5h_{xx}+|h|^2h=0$
>
> While initial conditions can prevent trivial solutions near ($t=0$), for long-term predictions, the optimization dynamics at larger times remain inaccurate (with a tendency for solutions to collapse, as described in previous works).
>
> This has motivated progressive training approaches that segment the time domain and iteratively propagate the initial conditions across subintervals. However, these methods have notable drawbacks: they require manual segmentation of the domain, adjustment of the training process, and are prone to error accumulation.
>
> Inspired by FPNN, we wonder if a mathematical transformation (e.g., exponential or logarithmic operations, such as $\log p$ in the FP equations) could equivalently reformulate the PDE and its associated loss function, such that the zero solution no longer satisfies the transformed equation. Could this ensure a coherent global optimization dynamic? This is merely an idea and a preliminary thought, requiring further research and experimental validation.
>
> $\textbf{Weakness 1}$
>
> Some limitations and discussions are addressed in the **Partition Function** section and **Theorem 1 in Appendix A**. These relate to potential failure scenarios of the two methods for estimating $Z_\theta$ (Eq.(8) and Eq.(10)) in higher dimensions, as well as issues such as NaN errors for $\log p$ in the loss function (e.g., for 6D Unimodal on $\Omega=[-2,2]^6$, where the true solution is exactly zero at the boundary in Python). We will revise the paper to present these aspects more clearly in the discussion section.
>
> $\textbf{Weakness 2}$
>
> Our computational complexity is significantly reduced compared to existing methods, because the normalization condition does not need to be explicitly considered during training. For instance, TFFN requires estimating $Z_\theta$ at every iteration, "soft" PINN necessitates computing $\mathcal{J}\_{\text{norm}}$ at each step, and density estimation with normalizing flows requires simultaneous tracking of $x$ and log-density changes. For PDE residuals, we only perform an additional $\log$ operation on the network output, and the computational costs for $\mathcal{J}\_{\text{plain}}$, $\mathcal{J}\_{\text{score}}$ and their computational graphs remain roughly comparable. Ultimately, we estimate $Z_\theta$ only once, making computational complexity entirely acceptable.
>
> If you have any more questions, please feel free to give comments or remarks.

---

> > ### Comment · Reviewer_aoGx · 2024-11-26
> >
> > thank you for the response.

---

### Official Review · Reviewer_1JZg · 2024-11-03

**Soundness:** 2
**Presentation:** 3
**Contribution:** 2
**Rating:** 5
**Confidence:** 2

**Summary:**

In this paper, the authors focus on steady state Fokker-Planck (SFP) equations and propose novel score-based PDE loss. The proposed loss does only depend on the score function $s_\theta$, avoiding the necessity to compute the normalization constant of the probability distribution. Furthermore, the authors propose to investigate the proposed loss on two types of architectures, namely tensor neural networks (TNNs) and MLPs. Experiments on several PDE examples are performed, showing the good performance of the proposed method.

**Strengths:**

1. The paper is overall well written and easy to follow.
2. Experimental results are interesting and support the authors' claims.

**Weaknesses:**

1. There is a potential clash with the notion of score in the ML/DL community, see questions.
2. It is difficult to evaluate the novelty of the work. For example, authors only compare the proposed approach to TFFN but no other baselines are provided. I would strongly suggest that the authors add more baselines to allow for an easier comparison with existing approaches, e.g. [1,3].
3. Similarly, it is not clear whether the set of chosen experiments are commonly used in the  PINN community. For example, is the dataset of the authors present in [2]?

(see refs in questions)

**Questions:**

**Main comments**
1. I am afraid that the notion of "score" that is central to the paper is fairly different to the one usually referred to in deep learning, where the score is implicitly learned through denoising and the associated distribution is never explicitely computed. Here, the authors rather use a differentiable network and use this property within the loss. Can the authors comment on that?
2. Why is the computation of $Z_\theta$ important in the context of Fokker-Planck (FP)? If I understand well, the authors parametrize FP with a neural network, with a direct access to both $\nabla \log p$ and $p$. Is the summation approach of (8) and (10) tractable as dimension increase?
3. I struggle to understand the point of the authors lines 453-473. Firstly, the notation $|D_{norm}|$ is slightly confusing. Secondly, what is the source of randomness in $D_{norm}$  mentionned by the authors line 465 ? If the dataset for $\mathcal{D}_{norm}$ is simulated, could the authors generate more samples?
4. Fig. 9 is slightly unclear. The plots on top and bottom show two different things (top: MAP and MAPE, bottom: MAPE and Z) for two different architectures. In particular, why should Z and the MAPE be related? Could the authors comment on that?

**Minor comments**
1. While the paper is well written, some typos are remaining, e.g. "Score-based generate model" (line 129)
2. The color scheme from Fig. 4 (c) (bottom) is not clear, it seems most of the maps are identically 0. The authors may want to reduce the threshold.
3. The message would be more striking in Fig. 5 and 6 if experiment was run multiple time with different random seeds. This would allow the authors to provide smoother curves with mean and error bars.


**References**

[1]
@inproceedings{zhai2022deep,
  title={A deep learning method for solving Fokker-Planck equations},
  author={Zhai, Jiayu and Dobson, Matthew and Li, Yao},
  booktitle={Mathematical and scientific machine learning},
  pages={568--597},
  year={2022},
  organization={PMLR}
}

[2]
@article{lu2021deepxde,
  author  = {Lu, Lu and Meng, Xuhui and Mao, Zhiping and Karniadakis, George Em},
  title   = {{DeepXDE}: A deep learning library for solving differential equations},
  journal = {SIAM Review},
  volume  = {63},
  number  = {1},
  pages   = {208-228},
  year    = {2021},
  doi     = {10.1137/19M1274067}
}


[3]
@article{cho2024separable,
  title={Separable physics-informed neural networks},
  author={Cho, Junwoo and Nam, Seungtae and Yang, Hyunmo and Yun, Seok-Bae and Hong, Youngjoon and Park, Eunbyung},
  journal={Advances in Neural Information Processing Systems},
  volume={36},
  year={2024}
}

---

> ### Author Response · Authors · 2024-11-18
>
> We sincerely appreciate your comments and constructive suggestions. In our **Official Comments**, we further elaborate on the distinctions and improvements of our proposed FPNN over previous works, which we believe will facilitate a better understanding of the PDE task and our method.
>
> **$\textbf{Weakness 1 and Question 1}$**
>
> **Our Task**
>
> Our goal is to solve an important class of PDE systems with boundary conditions: the steady-state Fokker-Planck (SFP) equations. Unlike other PDEs, the solution to the SFP equation represents the density function of an invariant equilibrium distribution, and needs to satisfy additional normalization constraints.
>
> Due to the high dimensionality, we want to leverage deep learning methods to find the solution to a given SFP equation, where the drift $\mu(x)$ and diffusion $D(x)$ are known in advance. It means that we can know which steady state the system reaches under both drift and diffusion effects.
>
> **Score Matching**
>
> In the **Our Work** section of Official Comments, we introduce the score PDE loss $\mathcal{J}\_{\text{score}}(s_\theta)$ and show that, when $D(x) = I_d$, the training process is equivalent to minimizing:
>
> $\mathbb{E}\_{p_\text{data}}[\|s_\theta(x)-\mu(x)\|^2]$
>
> In Song's work [10], the Noise Conditional Score Network (NCSN) uses a weighted sum of denoising score matching objectives:
>
> $\theta^*=\arg \min_{\theta}\sum_{i=1}^N \sigma_i^2\mathbb{E}\_{p_\text{data}(x)}\mathbb{E}\_{p_{\sigma_i}(\widetilde{x}|x)}[\|s_\theta(\widetilde{x},\sigma_i)-\nabla_{\widetilde{x}}\log p_{\sigma_i}(\widetilde{x}|x)\|_2^2] \qquad(1)$
>
> and Denoising Diffusion Probabilistic Model (DDPM) leverages a re-weighted variant of the evidence lower bound (ELBO):
>
> $\theta^*=\arg \min_{\theta}\sum_{i=1}^N (1-\alpha_i)\mathbb{E}\_{p_\text{data}(x)}\mathbb{E}\_{p_{\alpha_i}(\widetilde{x}|x)}[\|s_\theta(\widetilde{x},i)-\nabla_{\widetilde{x}}\log p_{\alpha_i}(\widetilde{x}|x)\|_2^2] \qquad(3)$
>
> It can be seen that our equivalent score-matching objective:
>
> $\theta^*=\arg \min_{\theta}\mathbb{E}\_{p_\text{data}(x)}[\|s_\theta(x)-\mu(x)\|^2]$
>
> shares a similar form with score-based generative models. These generative models gradually learn the "score" of the perturbed data distribution (i.e., conditional Gaussian distribution), while our FPNN directly learns the known drift $\mu(x)$.
>
> A notable difference is that generative models only need to accurately match the score to generate samples using Langevin MCMC or the estimated reverse Markov chain. But our goal is to compute the solution of SFP equation, which requires obtaining a physically meaningful normalized density. It necessitates the post-processing of calculating the partition function $Z_\theta$.
>
> **References**
>
> [10] @article{song2020score,
>   title={Score-based generative modeling through stochastic differential equations},
>   author={Song, Yang and Sohl-Dickstein, Jascha and Kingma, Diederik P and Kumar, Abhishek and Ermon, Stefano and Poole, Ben},
>   journal={arXiv preprint arXiv:2011.13456},
>   year={2020}
> }
>
> **$\textbf{Weaknesses 2 and 3}$**
>
> In the **Baselines** and **PDE Cases** sections of Official Comments, we provide a detailed comparison with existing works and conduct relevant experiments.
>
> **Ref [1]**
>
> As we can see, the FP solver [1] relies heavily on a large number of reference solutions, although they may not be exact and affect model training performance. Our FPNN does not require any labeled data and results in significant improvements in both efficiency and accuracy over the FP solver.
>
> **Ref [3]**
>
> SPINN [3] does not incorporate specific modifications for FP equations and faces challenges as we discussed in **Background**. Moreover, SPINN only demonstrates PDE examples up to 6 dimensions. Its tensor-product output will be storage infeasible in much higher-dimensional problems, because we need to expand the predicted density $p$ of size $N^d$ as an intermediate quantity, and then calculate the PDE residuals through automatic differentiation.
>
> Therefore, although our method uses the TNN architecture (same as SPINN), we adopt the original input-output format: the input has the shape (batch size, dim), while the output are densities with the shape (batch size, 1).
>
> **Ref [2]**
>
> Consider the challenges of solving high-dimensional SFP equations, we opted to evaluate our model using PDEs with analytical solutions. The 4D Ring is based on [1], while the 6D examples and 10D Multi-modal are adapted from [5]. Additionally, we construct cases involving the 10D Gaussian mixture distribution and 20D Gaussian Gaussian function to comprehensively test the applicability of our algorithm. FPNN consistently shows strong performance across these cases. Since our focus is on SFP equations and efficiently handling normalization constraints in a more elegant manner, the datasets used in our work are not covered in [2].

---

> > ### Author Response · Authors · 2024-11-18
> >
> > **$\textbf{Question 2}$**
> >
> > In **Question 1**, we explain that in order to obtain a physically meaningful solution, it is necessary to compute $Z_\theta$ once and perform normalization. Our neural network directly models the unnormalized density $\widetilde{p_\theta}$ and the score function $s_\theta = \nabla \log \widetilde{p_\theta}$. Since we observe that $\nabla \log \widetilde{p_\theta} = \nabla \log p_\theta$, we can use $s_\theta$ to approximate the true score.
> >
> > Your focus on Equations (8) and (10) is both correct and important. Firstly, FPNN is fundamentally different from existing methods like "soft" or "hard" PINNs. In our loss function, we do not require the evaluation of $Z_\theta$, as we completely decouple the normalization condition. This allows us to compute the partition function $Z_\theta$ only once after the model has been fully trained, unlike existing methods [4, 5], which necessitate re-estimating $Z_\theta$ at every training iteration.
> >
> > **TNN-based FPNN**
> >
> > For the TNN-based FPNN, efficient numerical integration is possible due to the low-rank structure of the density representation. For example, when $r = 1$, Eq.(8) simplifies to:
> >
> > $Z_\theta\approx(\int_{a_1}^{b_1}f_1(x_1)dx_1)\cdots(\int_{a_d}^{b_d}f_1(x_d)dx_d)$
> >
> > which means that the high-dimensional integral can be decomposed into a product of $d$ one-dimensional integrals (where $d$ is the spatial dimension). Each of these one-dimensional integrals can be efficiently estimated using any numerical algorithms (we use the piece-wise Gauss-egendre quadrature rule). As a result, even as the dimensionality increases, $Z_\theta$ remains computationally tractable. Furthermore, for $r$ components, the above calculations are fully consistent and parallelizable, as our code implementation.
> >
> > **MLP-based FPNN**
> >
> > The estimation in Eq.(8) essentially forms a high-dimensional grid using tensor products, which imposes certain requirements on the network architecture. Therefore, more generally, we provide a Monte Carlo estimation for the partition function, as described in Eq.(10). This approach does not suffer from the curse of dimensionality as grid-based numerical methods. Moreover, we can **increase the number of sampling points** to improve the accuracy of this unbiased estimator. Since this calculation is performed only once, the computational cost remains controllable.
> >
> > **$\textbf{Questions 3 and 4}$**
> >
> > Here, $|\mathcal{D}\_{\text{norm}}|$ is consistent with its usage in Eq.(10), representing the number of Monte Carlo (MC) samples, which are uniformly drawn from the domain $\Omega$. In Figure 9(a), our target is to examine how varying the number of MC samples affects the estimation of $Z_\theta$, and subsequently, the prediction error of the solution function. The experimental results demonstrate that increasing the number of uniformly sampled points $|\mathcal{D}_{\text{norm}}|$ indeed enhances the accuracy of the solution.
> >
> > **Comparison of Estimation Methods**
> >
> > For the TNN-based FPNN, in addition to using Eq.(8), we can also apply Eq.(10) to compute the partition function $Z_\theta$. We want to compare the difference and efficiency between these two methods. In Figure 9(b), the rightmost column displays the values of $Z_\theta$ estimated using Eq.(8) and the corresponding MAPE. We observe that, for the MC estimation in Eq.(10), as the number of uniform sampling points $|\mathcal{D}\_{\text{norm}}|$ increases, $Z_{\text{MC}}$ gradually converges to $Z_\theta$, and MAPE steadily decreases until stable.
> >
> > **Limitations of Higher-Dimensional Problems**
> >
> > In lines 453-458, we analyze the advantages and limitations of using Eq.(8) and (10) in particularly high-dimensional settings. Suppose the domain of interest is $\Omega = [-2, 2]^d$. When using Eq.(10) for estimation, the volume of $\Omega$ is $|\Omega| = 4^d$, which may exceed machine precision limits when $d = 20$, $40$, or higher dimensions. This issue is exacerbated when the range of each interval in $\Omega$ is larger, such as $[-2, 5.5]$ or $[-4, 4]$.
> >
> > In contrast, Eq.(8) involves computing the integral over each interval $(\int_{-2}^{2} f_i(x_i)dx_i)$ separately, followed by multiplying these values to obtain $Z_\theta$. In this case, the integral value for each dimension can be small or close to 1. Even if the range of each interval is larger, the integral values remain stable, making this approach more robust to dimensionality increases.

---

> > > ### Author Response · Authors · 2024-11-18
> > >
> > > **$\textbf{Minor comments}$**
> > >
> > > 1. Thanks for your comment. We will carefully review our manuscript for spelling and grammatical issues, and we will clarify the notation and descriptions in lines 453-473 to enhance readability. Your feedback is invaluable in improving the clarity of our work.
> > >
> > > 2. Figure 4 primarily compares the efficiency between TFFN and FPNN. We illustrate that FPNN is able to produce accurate predictions within fewer than 1,000 iterations, whereas TFFN still exhibits high errors with 1,000 iterations and fails to converge properly even after 10,000 iterations (steps). This highlights the significant advantage of the loss function $\mathcal{J}\_{\text{score}}$ over $\mathcal{J}\_{\text{plain}}$. In the second row, both models utilize the same error colorbar for fair comparison.
> > >
> > > 3. Your suggestion is much appreciated. Currently, we have set a consistent random seed of **"111"** across all experiments to ensure reproducibility of our results. FPNN consistently aligns well with the ground truth solutions for various high-dimensional SFP equations. And we are confident that the superior performance of FPNN is robust and not coincidental, as our improvements to the loss function are fundamental and reasonable.

---

> > > > ### Comment · Reviewer_1JZg · 2024-11-24
> > > >
> > > > I thank the authors for their detailed response which answers some of my concerns. Please find below one last point where I am unsure I understand correctly the authors.
> > > >
> > > > **Question 1.** From the generative modeling perspective, the authors are precisely learning the drift term, which is not related to the score. Is my understanding correct? If yes, while learning the drift (in the authors' context) may be interesting, I find that the formulation as score matching may be more confusing than helping the reader.

---

> > > > > ### Author Response · Authors · 2024-11-25
> > > > >
> > > > > Thank you for your comments. Indeed, our approach involves learning the drift term to obtain the density function. But the drift term and the score are not unrelated. Instead, it is closely related to the concept of sampling in score-based generative models.
> > > > >
> > > > > The figure (https://postimg.cc/75BxMCBj) illustrates this connection. We present score-based generative modeling (Fig.(a)) and plot the vector field of the drift term $\mu(x)$ for our 4D Ring and 6D Multi-modal problems (Fig.(b)). It can be observed that the "drift" term essentially describes the "score", which corresponds to the desired sampling direction.
> > > > >
> > > > > **One more idea**
> > > > >
> > > > > Song's work [10] show that the noise perturbations of SMLD and DDPM are discretizations of the Variance Exploding (VE) and Variance Preserving (VP) SDEs, respectively.
> > > > >
> > > > > **VE SDE:** $dx=\sqrt{\frac{d[\sigma^2(t)]}{dt}}dw_t\quad(9)$
> > > > >
> > > > > **VP SDE:** $dx=-\frac{1}{2}\beta(t)xdt+\sqrt{(\beta(t))}dw_t\quad(11)$
> > > > >
> > > > > These ideas are both natural and elegant. To progressively perturb the true data distribution into a (standard) Gaussian distribution, infinite variance would be required in the absence of a drift term ($\mu(x)=0$). Alternatively, by introducing a drift term $\mu(x) = -\frac{1}{2}\beta(t)x$ toward the origin, this transformation can be achieved while preserving the variance. (This drift term $\mu = -ax$ is used in our 20D Gaussian problem.)
> > > > >
> > > > > Thus, an interesting question arises: could more complex drift terms be designed to characterize perturbative SDEs (e.g., to control image classes) and obtain other desirable properties? This is a potential topic for further exploration. However, as *Image Generation* is not my research focus, my understanding may be limited.
> > > > >
> > > > > I hope our response helps to address your questions. Your comments and questions are crucial for summarizing our work and have prompted us to reflect on the connection between the score PDE loss and score matching. These conclusions will be included in Appendix A.

---

> > > > > > ### Author Response · Authors · 2024-11-27
> > > > > >
> > > > > > We have thoroughly revised the manuscript based on your suggestions and comments. The specific changes (highlighted) are as follows:
> > > > > >
> > > > > > 1. In the **introduction**, we have provided a detailed explanation of the differences and connections between our work and previous advanced studies.
> > > > > > 2. The **partition function** section has been revised, including the experimental descriptions and formula representations.
> > > > > > 3. We have added supplementary material to clarify the relationship between score PDE loss and score matching.
> > > > > > 4. New figures have been included to illustrate the MAE of  our approximate solution. In addition to the numerical results in tables, these figures visually show that FPNN achieves an order-of-magnitude reduction in MAE compared to TFFN across all tested SFP equations, while also facilitating an intuitive comparison with other related works, such as ref [1].
> > > > > >
> > > > > > Considering these improvements, we sincerely invite you to review our work and hope it can inspire greater confidence in your assessments. Thanks for your time and consideration! And thank you for all reviewers.

---

> > > > > > > ### Comment · Reviewer_1JZg · 2024-12-02
> > > > > > >
> > > > > > > I thank the authors once again for taking the time to reply. I agree with the authors that the drift and score terms are related in score matching, however the link with score matching is still unclear ; in the authors response, the sentence "It can be observed that the "drift" term essentially describes the "score", which corresponds to the desired sampling direction." is particularly confusing.
> > > > > > >
> > > > > > > 1. If I understand correctly, the authors suggest that in the specific context of FP, solving the problem with score matching amounts to learning the drift coefficient of the PDE. In this case, the paper needs to be clarified and the addition of the authors with Fig. 11 is not going in the correct direction since it conveys the message that both drift and score play the same role ("both represent gradient information of the data distribution", which is not true). I would rather suggest that the authors clearly acknowledge the difference between the two views.
> > > > > > >
> > > > > > > 2. It is not clear for me what is gained overall from the PINN approach for getting samples of the underlying distribution since it can be simulated from the physical equations. Could the authors please clarify on this point?
> > > > > > >
> > > > > > > Based

---

> > > > > > > > ### Author Response · Authors · 2024-12-02
> > > > > > > >
> > > > > > > > Thank you for thoughtful comments again, which make your questions or concerns clear to us.
> > > > > > > >
> > > > > > > > **Question 2**
> > > > > > > >
> > > > > > > > Allow us to address the second point first. The purpose of employing PINNs to solve SFP equations is not (or not merely) to obtain samples but to estimate the probability density. This is also why normalizing flows (such as KRnet) can be used. These architectures leverage the change of variables formula to estimate the log-density, thereby providing the density values. If our goal is only to generate samples, then the SDE simulation is sufficient, as in generative models.
> > > > > > > >
> > > > > > > > The estimation of probability density plays a crucial role in anomaly detection or fault diagnosis in stochastic systems. With access to the response probability density, we can easily determine whether a particular state or feature of the system ($x_0\in\mathbb{R}^d$) is an outlier or a low-probability event (e.g., $p(x_0)<\epsilon$). This highlights the importance of normalized and "standard" probability densities in our analysis. For unnormalized densities $\widetilde{p_\theta}$, their absolute values offer limited utility, as they only allow relative comparisons without providing intuitive insights into the likelihood of specific events.
> > > > > > > >
> > > > > > > > **Question 1**
> > > > > > > >
> > > > > > > > Thanks for your first question and suggestions, which provides us with an opportunity to elaborate further. Our main points are summarized as follows:
> > > > > > > >
> > > > > > > > - When improving the PINN loss function, we utilize the transformation formula $s_\theta=\nabla\log \widetilde{p_\theta}$, thereby incorporating the concept of "score".
> > > > > > > > - "Drift," as a physical concept, is used in both SDEs and FP equations. Constructing drift terms involving "score" and reverse-time SDEs is fundamental for generating data and sampling images.
> > > > > > > >
> > > > > > > > Below, we explain the second point. In Song's work [10] (It is a remarkable study that balances mathematical elegance and practical utility and we deeply respect), we consider three distinct dynamical equations:
> > > > > > > >
> > > > > > > >
> > > > > > > >
> > > > > > > > 1. For the **SDE:** $dx=f(x,t)dt+G(x,t)dW\quad(1)$,
> > > > > > > >
> > > > > > > >    its PDF $p(x,t)$ satisfies the FP equation: $\frac{\partial p}{\partial t}=-\nabla\cdot(fp)+\nabla\cdot\nabla\cdot(\frac{1}{2}GG^Tp)\quad(2)$.
> > > > > > > > 2. For the **probability flow ODE:** $dx=(f(x,t)-\frac{1}{2}GG^T\nabla\log p)dt\quad(3)$,
> > > > > > > >
> > > > > > > >    substituting the drift and (zero) diffusion terms into the FP equation yields: $\frac{\partial p}{\partial t}=-\nabla\cdot((f(x,t)-\frac{1}{2}GG^T\nabla\log p)p)=-\nabla\cdot(fp)+\nabla\cdot\nabla\cdot(\frac{1}{2}GG^Tp)$, which is fully consistent with Eq.(2) (assuming $G(t)$ is independent of $x$). Since Eq.(3) does not involve a diffusion term, ODE is reversible, allowing denoising processes to be carried out effectively.
> > > > > > > > 3. For the **reverse-time SDE:** $dx=(f(x,t)-GG^T\nabla\log p)dt+G(x,t)dW\quad(4)$,
> > > > > > > >
> > > > > > > >    the density $p$ is still governed by Eq.(2), though its corresponding FP equation now describes the reverse-time solution. Compared to Eq.(3), SDE in Eq.(4) provides additional randomness needed to sample from the true distribution, enabling the generation of new samples.
> > > > > > > >
> > > > > > > > When we set the drift term $f(x,t)=-\frac{1}{2}\beta(t)x$, any real image under the influence of the drift tends toward the origin. Under the stochastic process in Eq.(1), the data distribution converges to a standard Gaussian distribution $p_G$. This connection with $p_G$ facilitates subsequent sampling random noise from $p_G$ for generation.
> > > > > > > >
> > > > > > > > Using reverse diffusion sampling based on Eq.(4) as an example, the new drift term is $\widetilde{f}(x,t)=f(x,t)-GG^T\nabla\log p$, where the first term reverses the process of toward the origin, and the second term guides to the data distribution. Both terms are conditioned on the current state $x$.
> > > > > > > >
> > > > > > > > Thus, a clearer explanation would be: we construct the drift ($\widetilde{f}$) with the score ($\nabla \log p$), which contains the gradient information of data distribution and plays a dominant role. This ultimately reflects the consistency between the score and drift, as both indicate the direction of the true data distribution.
> > > > > > > >
> > > > > > > > From the perspective of PDE systems, our focus is on the forward problem, solving the known equations to obtain solutions. In contrast, the training process of generative models can be viewed as the inverse problem, where the goal is to infer equation coefficients or unknown terms from data.
> > > > > > > >
> > > > > > > > We hope this response clarifies your concerns! If time permits, we look forward to further discussions with you, as this would greatly help us refine the clarity of our manuscript. Thanks for your comments again.

---

> > > > > > > > > ### Author Response · Authors · 2024-12-03
> > > > > > > > >
> > > > > > > > > Here, we summarize the tasks of image generation and SFP equations:
> > > > > > > > >
> > > > > > > > > - **Generative Models:** For score-based generative models, Eq.(1) describes the forward process of adding noise, which cannot be used for sampling images. Image generation relies on Eq.(3) or Eq.(4): $dx=(f(x,t)-GG^T\nabla\log p)dt+G(x,t)dW$.
> > > > > > > > >
> > > > > > > > >   This SDE contains an unknown term, $\nabla\log p$, which needs to be inferred. The goal is to map a standard Gaussian distribution to the real data distribution. Since the dynamics are oriented to data distribution, the model can only learn in a data-driven approach.
> > > > > > > > >
> > > > > > > > > - **FPNN:** Solving the SFP equation corresponds to a known SDE: $dx=f(x)dt+G(x)dW$.
> > > > > > > > >
> > > > > > > > >   PINNs solve it by directly minimizing the PDE residual during training. Our method leverages the equivalence between Eq.(1) and Eq.(3), reformulating the problem into an SDE: $dx=(f(x)-\frac{1}{2}GG^T\nabla\log p)dt=\widetilde{f}(x)dt$.
> > > > > > > > >
> > > > > > > > >   This step introduces the score function associated with the solution of FP equation, where $f(x)=\mu(x),\frac{1}{2}GG^T=D(x)$. Due to the stationary invariant property, we set $\widetilde{f}(x)=0$ to train the network.
> > > > > > > > >
> > > > > > > > > - Although the two methods exhibit formal similarities, the generative model has a nonzero $\widetilde{f}(x,t)$ and drives towards the data distribution, while FPNN maintains a steady state by setting $\widetilde{f}(x)=0$. \
> > > > > > > > > The former is data-driven, whereas the latter is guided by physical laws.
> > > > > > > > >
> > > > > > > > > We hope our explanation is helpful to you!

---

### Author Response · Authors · 2024-11-17

We greatly appreciate all the reviewers’ comments and constructive suggestions, which are extremely helpful in better organizing and presenting our work. Please allow us to first provide the background of our study, reiterate our motivation and improvements. Then we will respond to the concerns about comparisons and experiments with related work. This will help reviewers understand our algorithm and better evaluate our contributions.

$\textbf{Background}$

PINN is a general deep-learning framework for solving PDEs and has achieved significant success in various problems, such as the Navier-Stokes equations, Allen-Cahn equation, Schrödinger equation, etc. However, PINNs face challenges in Fokker-Planck equations, with the objective function: (plain PDE loss)

$\mathcal{J}\_{\text{plain}}(p_\theta)=\mathbb{E}\left[|\nabla\cdot(p_\theta\mu) - \nabla \cdot(\nabla\cdot(Dp_\theta))|\right]$

It is evident that the zero solution $p_\theta = 0$ also satisfies this equation. Therefore, directly minimizing the loss function above often leads to network collapse, where $p_\theta$ converges to a trivial solution. Existing strategies generally fall into two categories involving normalization constraints in soft or hard manner:

1. Adding a penalty term $\mathcal{J}\_{\text{norm}}=\left(\frac{|\Omega|}{|\mathcal{D}|}\sum_{x\in\mathcal{D}} p_\theta(x) - 1\right)^2$, where the dataset $\mathcal{D}$ is sampled uniformly from $\Omega$ [4].
2. Developing specialized structures to represent the density function [5-9].

However, the former method requires manual balancing of multi-objective losses, while the latter may limit the model's representation capacity (Figure 1).

Another earlier data-driven method guides the network to the desired solution by introducing reference solutions and the regression loss $\mathcal{J}\_{\text{label}}=\frac{1}{N^Y}\sum_{j=1}^{N^Y}(p_\theta(x_j)-p_j)^2$, i.e. **FP solver** [1] suggested by Reviewer 1JZg. Our 4D Ring problem is directly from this work.

For this case, FP solver utilizes a very large number of particles ($10^{10}$ sample points) for SDE simulations and constructs frequency histograms on a grid as reference solutions ($10^4$ reference points). This technique is computationally intensive and lacks scalability to higher dimensions.

As seen in Figure 6 of [1], despite using such a large amount of points, the $L_2$ error of FP solver is around $10^{-2}$. In contrast, our FPNN achieves a MAE of $5.56 \times 10^{-4}$ without any labeled data (resulting in an even lower $L_2$ error). Furthermore, FPNN reaches this superior performance with fewer parameters and only 2,000 unlabeled data points per iteration. FPNN requires fewer than 1,000 iterations to surpass this baseline performance (see Figure 4(a) in our work and Figure 6 in [1]).

$\textbf{Our work}$

Our method deviates from all existing approaches by changing the loss function directly. The improvements we have achieved can be attributed to a fundamental modification of the loss function, which decouples the normalization condition from the training process. Specifically, we replace the original loss term $\mathcal{J}_{\text{plain}}$ with the following:

$\mathcal{J}\_{\text{score}}(s_\theta)=\mathbb{E}\left[|\nabla\log \widetilde{p_\theta} \cdot \widetilde{\mu}(x) + \nabla\cdot\widetilde{\mu}(x)|\right], \quad \widetilde{\mu}(x) = \mu(x) - \nabla\cdot D(x) - D(x)\nabla\log \widetilde{p_\theta}$

In this form, the trivial solution $\widetilde{p_\theta}=0$ no longer satisfies the equation. The *score PDE loss* maintains equivalence with the *plain PDE loss* while avoiding a zero solution and eliminating the need for handling normalization conditions in training process.

To further illustrate, consider a special case where $D(x) = I_d$. If minimizing the score loss $\mathcal{J}_{\text{score}}$ drives $\widetilde{\mu}(x)$ to zero, the training process becomes equivalent to optimizing the objective function of *flow matching*:

$\mathbb{E}\left[\|\mu(x) - \nabla\log \widetilde{p_\theta}\|^2\right].
$

Thus, our method inherently integrates score matching while preserving the original Fokker-Planck equation and corresponding physical laws.

**Our main target** is to fully decouple the normalization condition during training, allowing the network to freely adjust the magnitude to learn the unnormalized density $\widetilde{p_\theta}$ (note that our network directly models $\widetilde{p_\theta}$ rather than $p_\theta(x)$ and appropriate scale is really necessary for learning high-dimensional densities). Subsequently, it is natural to perform a postprocess of calculating the partition function $Z_\theta$ under the score PDE loss. We derive the approximate solution as $p_\theta(x) = \widetilde{p_\theta}(x) / Z_\theta$. The **advantages** are also clear: it reduces computational costs, eliminates the interference caused by normalization constraints on optimization dynamics, and ensures a more efficient and coherent training process.

---

> ### Author Response · Authors · 2024-11-17
>
> $\textbf{Baselines}$
>
> In our theoretical analysis, we extensively reviewed and examined a range of state-of-the-art methods and baselines, identifying specific limitations that hinder their performance in higher-dimensional problems. This analysis forms the basis of our motivation to address these challenges.
>
> 1. **Data-driven methods** (e.g., [1]), as demonstrated, are inferior to our FPNN in terms of both efficiency and accuracy. There are no particularly accurate densities, and we still need to balance two loss terms $\mathcal{J}\_{\text{plain}}$ and $\mathcal{J}\_{\text{label}}$.
>
> 2. **"Soft" PINNs** are effective in lower-dimensional SFP equations, such as 2D Ring, but tend to fail as dimensionality increases. Despite extensive efforts to balance $\mathcal{J}\_{\text{plain}}$ and $\mathcal{J}\_{\text{norm}}$ using gradient norms, we could only marginally obtain the solution of 4D Ring, and even then, stable convergence was not achieved. In the 6D Multi-modal problem, this method failed entirely.
>
> 3. **"Hard" PINNs** often sacrifice model's representation capacity, and their optimization dynamics are not smooth. We believe this tortuous optimization process is a key factor limiting their scalability to higher dimensions.
>    The recent developed method, TFFN [5], introduced this year, first claimed to solve such high-dimensional as well as complex SFP equations. It bears the closest resemblance to our approach, making it a worthy baseline for comparison. (We would like to express our gratitude to the authors of this work, as our method was refined and matured through ongoing exploration and analysis of TFFN.)
>
>    In our experiments, both TFFN and our TNN-based FPNN use the same network architecture, spatial domain, number of training points, optimizer, learning rate, and other hyper-parameters. The only difference lies in **replacing $\mathcal{J}\_{\text{plain}}$ with $\mathcal{J}\_{\text{score}}$**, which can be seen as an ablation study to evaluate the effectiveness of our score-based FP loss.
>
>    As shown in Figures 6(b) and 6(c), TFFN exhibits noticeable deviations from the true solution in 6 dimensions, resulting in a MAPE of 293% and 92.90% (see Table 1). For the 10D Multi-modal case, TFFN fails to accurately identify the two density peaks. Thus, comparisons with TFFN are omitted for 10-20 dimensional examples.
>
> 4. Other general PDE solvers, such as **DeepXDE [2], SPINN [3], DeepONet, and FNO** do not incorporate specific modifications for the Fokker-Planck equation and face similar issues as discussed above.
>
> To summarize, while our experiments present part of comparisons, our theoretical analysis comprehensively covers existing deep learning methods for the Fokker-Planck equation. Our score loss are concise and effective, demonstrating substantial originality. To the best of our knowledge, few works can effectively solve such high-dimensional, complex and challenging SFP equations, which also makes it difficult to find suitable and comparable baselines. Our algorithm fills a significant gap in the field of high-dimensional Fokker-Planck equations and achieves breakthrough performance, both theoretically and experimentally.
>
> $\textbf{PDE Cases}$
>
> Our test cases are from multiple studies for solving FP equations. Existing methods cannot learn so many types of challenging high-dimensional density functions effectively. Therefore, we focus on exploring the full potential of our FPNN, evaluating its generality and applicability across diverse problems.
>
> - **SFP equations.** The experiments include ring-shape density, arbitrary potential functions (where the polynomial degree in the exponential term reaches up to 8, with complex interactions among spatial coordinates), and Gaussian mixture distribution (with scalability to more components and higher dimensions, allowing for modeling more complicated distributions). Across all these cases, our method demonstrated exceptional performance and potential.
>
> - **Test dataset.** For high-dimensional problems, we are limited to visualizing the results using selected cross-sections. However, only testing errors on cross-sectional data is not sufficient to characterize high-dimensional solutions due to their multi-modal complexity. Thus, we generated $\mathcal{D}\_{\text{test}}$ to globally evaluate error metrics.
>
>   For the test dataset $\mathcal{D}\_{\text{test}}$, we generate data via gradient ascent method on the analytical solution, ensuring that all densities of test data exceed a predefined threshold $\epsilon$. This approach is more efficient than the traditional method of randomly sampling spatial points and filtering out those with densities below $\epsilon$.
>
> We have provided the codes for generating test datasets and fixed the random seed for reproducibility. In addition, we plan to release the codes and test datasets in a GitHub repository, enabling future research to perform comparisons with FPNN under consistent evaluation metrics.

---

> > ### Author Response · Authors · 2024-11-17
> > **References in comments**
> >
> > **References**
> >
> > [1] @inproceedings{zhai2022deep, title={A deep learning method for solving Fokker-Planck equations}, author={Zhai, Jiayu and Dobson, Matthew and Li, Yao}, booktitle={Mathematical and scientific machine learning}, pages={568--597}, year={2022}, organization={PMLR} }
> >
> > [2] @article{lu2021deepxde, author = {Lu, Lu and Meng, Xuhui and Mao, Zhiping and Karniadakis, George Em}, title = {{DeepXDE}: A deep learning library for solving differential equations}, journal = {SIAM Review}, volume = {63}, number = {1}, pages = {208-228}, year = {2021}, doi = {10.1137/19M1274067} }
> >
> > [3] @article{cho2024separable, title={Separable physics-informed neural networks}, author={Cho, Junwoo and Nam, Seungtae and Yang, Hyunmo and Yun, Seok-Bae and Hong, Youngjoon and Park, Eunbyung}, journal={Advances in Neural Information Processing Systems}, volume={36}, year={2024} }
> >
> > [4] @article{alhussein2023physics,
> >   title={Physics-Informed Solution of The Stationary Fokker-Plank Equation for a Class of Nonlinear Dynamical Systems: An Evaluation Study},
> >   author={Alhussein, Hussam and Khasawneh, Mohammed and Daqaq, Mohammed F},
> >   journal={arXiv preprint arXiv:2309.16725},
> >   year={2023}
> > }
> >
> > [5] @article{wang2024tensor,
> >   title={Tensor neural networks for high-dimensional Fokker-Planck equations},
> >   author={Wang, Taorui and Hu, Zheyuan and Kawaguchi, Kenji and Zhang, Zhongqiang and Karniadakis, George Em},
> >   journal={arXiv preprint arXiv:2404.05615},
> >   year={2024}
> > }
> >
> > [6] @article{al2022extensions,
> >   title={Extensions of the deep Galerkin method},
> >   author={Al-Aradi, Ali and Correia, Adolfo and Jardim, Gabriel and de Freitas Naiff, Danilo and Saporito, Yuri},
> >   journal={Applied Mathematics and Computation},
> >   volume={430},
> >   pages={127287},
> >   year={2022},
> >   publisher={Elsevier}
> > }
> >
> > [7] @article{tang2022adaptive,
> >   title={Adaptive deep density approximation for Fokker-Planck equations},
> >   author={Tang, Kejun and Wan, Xiaoliang and Liao, Qifeng},
> >   journal={Journal of Computational Physics},
> >   volume={457},
> >   pages={111080},
> >   year={2022},
> >   publisher={Elsevier}
> > }
> >
> > [8] @article{feng2022solving,
> >   title={Solving Time Dependent Fokker-Planck Equations via Temporal Normalizing Flow},
> >   author={Feng, Xiaodong and Zeng, Li and Zhou, Tao},
> >   journal={Communications in Computational Physics},
> >   volume={32},
> >   number={2},
> >   pages={401--423},
> >   year={2022}
> > }
> >
> > [9] @article{anderson2024fisher,
> >   title={Fisher information and shape-morphing modes for solving the Fokker--Planck equation in higher dimensions},
> >   author={Anderson, William and Farazmand, Mohammad},
> >   journal={Applied Mathematics and Computation},
> >   volume={467},
> >   pages={128489},
> >   year={2024},
> >   publisher={Elsevier}
> > }

---

### Meta-Review · Area_Chair_7mPA · 2024-12-22

**Metareview:**

This paper proposes a "score Fokker-Planck neural net" to address the issue in trying to solve Fokker-Planck equations by PINNs. The idea is to first write the dynamics of the score function under Fokker-Planck and first solve this score Fokker-Planck equation. Next, normalization is computed numerically in an efficient way that derives from a tensorial parameterization of the neural net. I think this is a nice contribution to ML stochastic dynamical systems that warrants publication. There are some omissions in the bibliography (for example the score evolution from here: https://arxiv.org/abs/2210.04296), but nothing damning.

**Additional Comments On Reviewer Discussion:**

The longest discussion was with 1JZg who asked good explanatory questions (how is this connected to denoising, is it the same "score", how is score related to drift, ...) which I believe helped authors tune the narrative for the audience familiar with generative diffusion models but not so much with Fokker-Planck equations. aoGx was asking about practicality, scalability, and baselines. hrfk wondered about SRK as part of data generation, OOD performance, and additional baselines. eksf asked about fairness of comparison with PINNs given that normalization is done in post-processing. Authors have successfully addressed many of these concerns. There was overall a consensus that this is a solid contribution to modeling of stochastic dynamical systems.

---

### Decision · Program_Chairs · 2025-01-22

Accept (Poster)